ecology, health and disease and epidemiology

chiroptera, landscape structure, Texas, white-nose syndrome, source-sink dynamics, disease management

**Author for correspondence:**
Thomas M. Lilley
e-mail: thomas.lilley@helsinki.fi

# Ten-year projection of white-nose syndrome disease dynamics at the southern leading-edge of infection in North America

Melissa B. Meierhofer[1,2,3], Thomas M. Lilley[3], Lasse Ruokolainen[4], Joseph S. Johnson[5], Steven R. Parratt[6], Michael L. Morrison[1], Brian L. Pierce[2], Jonah W. Evans[7] and Jani Anttila[8]

[1]Department of Rangeland, Wildlife and Fisheries Management, and [2]Natural Resources Institute, Texas A&M University, 534 John Kimbrough Boulevard, College Station, TX 77843, USA
[3]Finnish Museum of Natural History, University of Helsinki, Pohjoinen Rautatiekatu 13, 00100 Helsinki, Finland
[4]Department of Biosciences, University of Helsinki, Yliopistonkatu 4, 00100 Helsinki, Finland
[5]Department of Biological Sciences, Ohio University, Athens, OH 45701, USA
[6]Department of Ecology and Evolution, University of Liverpool, Liverpool L69 7BE, UK
[7]Wildlife Diversity Program, Texas Parks and Wildlife, 4200 Smith School Road, Austin, TX 78744, USA
[8]Natural Resources Institute Finland (Luke), Latokartanonkaari 9, 00790 Helsinki, Finland

MBM, 0000-0003-2384-1999; TML, 0000-0001-5864-4958; LR, 0000-0003-0951-9100; JSJ, 0000-0003-2555-8142; SRP, 0000-0002-5801-876X; JA, 0000-0002-2102-1930

Predicting the emergence and spread of infectious diseases is critical for the effective conservation of biodiversity. White-nose syndrome (WNS), an emerging infectious disease of bats, has resulted in high mortality in eastern North America. Because the fungal causative agent *Pseudogymnoascus destructans* is constrained by temperature and humidity, spread dynamics may vary by geography. Environmental conditions in the southern part of the continent are different than the northeast, where disease dynamics are typically studied, making it difficult to predict how the disease will manifest. Herein, we modelled WNS pathogen spread in Texas based on cave densities and average dispersal distances of hosts, projecting these results out to 10 years. We parameterized a predictive model of WNS epidemiology and its effects on bat populations with observed cave environmental data. Our model suggests that bat populations in northern Texas will be more affected by WNS mortality than southern Texas. As such, we recommend prioritizing the preservation of large overwintering colonies of bats in north Texas through management actions. Our model illustrates that infectious disease spread and infectious disease severity can become uncoupled over a gradient of environmental variation and highlight the importance of understanding host, pathogen and environmental conditions across a breadth of environments.

## 1. Introduction

Emerging infectious diseases of wildlife are increasing in number and threatening several species with extinction [1–3]. Emerging infectious diseases are those newly appearing or rapidly increasing in a population [4], occurring when pathogenic or putatively pathogenic organisms in the environment have the opportunity to infect new hosts species or populations. Changing environmental conditions can accelerate this process of host-switching by driving changes in host-species' distributions and by creating new habitat for pathogens found in environmental reservoirs [5,6]. The spread of these diseases is mediated by differences in host ecology and physiology, resulting in various

Proc. R. Soc. B 288: 20210719

patterns of spatial spread [7–9]. Therefore, predicting the spatial structure of future emerging infectious disease epidemics requires integration of both environmental factors and species-specific ecology and behaviour that can underpin pathogen contact networks [10].

An emerging infectious disease of bats known as white-nose syndrome (WNS) threatens the survival of populations of several cave-hibernating species in North America [11]. Since it was first documented, the fungal causative agent *Pseudogymnoascus destructans* has spread across North America at a rate of 200 to 900 km per year and is associated with host mortality in excess of 90% [11,12]. Although, bat-to-bat transmission is the primary mode of disease dispersal [13], *P. destructans* can persist in an environment devoid of bats [14]. The disease disrupts hibernation behaviour through multiple pathways [15–17] leading to an increased arousal frequency and ultimately, the depletion of fat reserves [18]. This has generated predictions of local extirpations and extinctions of once common bat species [19–21]. There is therefore a need to understand future spread so that conservation efforts can be prioritized. Moving towards this understanding will require understanding how factors associated with WNS transmission work together to influence spread.

The vegetative growth of *P. destructans* is constrained by temperature and humidity inside hibernacula [22,23] while the spread of the fungus is influenced by internal and external factors. Factors known to be associated with fungal transmission include bat species composition and abundance, population demographics [24], geography (e.g. distribution, frequency and connectivity of hibernacula) and climate [25–27]. Thus, the fungal spread may vary by geography and demography. Predictive modelling of WNS has focused on data collected from the northeastern United States (e.g. [26,28]). Consequently, findings from these studies may not reflect regional differences among bat hibernacula [28–30]. It is therefore important to understand the incidence—and prevalence of—WNS over different spatial and temporal scales to determine the potential impacts of disease [31].

Texas provides a unique situation for studying disease spread as it has the greatest number of bat species of any state in the United States [32]. *Pseudogymnoascus destructans* was first detected in north Texas in 2017 [33], with WNS first identified on cave myotis (*Myotis velifer*) in central Texas in 2020 [34]. However, it is unclear if environmental conditions in Texas caves [35], or their spatial distribution and frequency [27], are favourable for the persistence of *P. destructans* in Texas. Unlike in northeastern North America, there has not yet been substantial mortality documented or reported resulting from WNS in Texas. Owing to declines documented in other regions of North America, researchers are currently deploying treatments in Texas hibernacula to prevent pathogen exposure and reduce disease severity. Thus, understanding whether WNS can develop in the cave network in Texas and how the disease may move throughout the southern region is integral in implementing proper management strategies for caves.

Here, we used averaged bat demographics of hibernating species (*Eptesicus fuscus*, *M. velifer* and *Perimyotis subflavus*) and cave environmental data collected at the leading edge of WNS pathogen spread in Texas to develop a general model that captures a wider geographical range and conditions. For values pertaining to infection and recovery rate

from WNS, we integrated bat demographics of the little brown bat (*Myotis lucifugus*), a hibernating bat species not known in Texas, as minimal data existed for other species known in Texas. We model the probability of *P. destructans* being able to infect hosts, leading to symptoms of WNS, and furthermore, the death of the host. Herein, we hypothesized that: (i) spread is accelerated by high concentrations of caves and bat abundance across the landscape; and (ii) disease development is hindered by internal and external environmental conditions affecting both bat physiology and fungal growth. We predicted that: (i) spread will accelerate in central Texas; and (ii) north Texas will support disease development with only some sites with environmental characteristics conducive to WNS development in central Texas. We projected our results 10 years ahead to provide stakeholders information on how the disease will most likely behave to better implement conservation measures.

## 2. Material and methods

### (a) Model development

Our model is a modification of the patch model published by Lilley *et al.* [27] (full model description in the electronic supplementary material). In comparison with the previously published model, we have simplified the hibernation and transmission dynamics to achieve easier parameterization, and do not consider environmental stochasticity. Although many factors can affect temperatures of caves [36], and thereby affect bat abundance, these data were not readily available and their inclusion would have further complicated the interpretation of the model results. Our model consists of differential equations, with a periodic temperature forcing, describing the dynamics of the bat hosts and the free pathogenic fungus. We divided the hosts into susceptible, exposed and infectious, all of which can be either active or hibernating, leading to seven compartments in total. The dynamic state is tracked in a network of patches representing caves within the counties of Texas. We implemented in C++ and full program codes are available at https://github.com/janivaltteri/wnstexas.

We used a simple linear force of infection in response to environmental fungal density instead of the sigmoidal response used in the model of Lilley *et al.* [27]. Additionally, we used simple threshold functions for bat population growth rate and the transfer rates between active and hibernation states. While the original formulation in Lilley *et al.* [27] is theoretically sound and results in smooth dynamics, our current formation is analytically more tractable/computationally better suited to integrating real-world variation in parameters. The sigmoidal infectivity response has, however, notable effects on disease dynamics. Therefore, we replicated all simulation experiments using different sigmoidal parameterizations and show the resulting effects in the electronic supplementary material.

Hibernation strongly affects disease dynamics because *P. destructans* has different effects on active and hibernating bats [37]. Thus, we determined the duration of hibernation in a patch by ambient and hibernaculum temperatures, $T_{amb}$ and $T_{hib}$, together with three threshold values $\alpha$. A patch is in hibernation when either $T_{amb}(t) < \alpha_{amb,0}$ or $T_{amb}(t) < \alpha_{amb,1}$; $T_{hib}(t) < \alpha_{hib}$. Our parameter set used threshold values $\alpha_{amb,0}$, $\alpha_{hib} = 11.5°C$ and $\alpha_{amb,1} = 12.5°C$ [35,36].

### (b) Spatial setting

In Texas where greater than 95% of the land is privately owned, caves, as opposed to other hibernacula (e.g. culverts), are challenging to monitor and manage for WNS because of access

restrictions. Despite access difficulties, we focused on caves for model development to assist with identifying regions to focus access efforts for future monitoring and management. Additionally, we chose caves because of the lack of available data on environmental characteristics of alternative hibernacula, and to retain the simplicity of the model.

We obtained information on the number of caves per county within Texas from the Texas Speleological Survey (https://www.texasspeleologicalsurvey.org/). The Texas Speleological Survey, Texas Cave Management Association, local Grottos, biologists and private landowners provided access to cave sites for data collection.

We gathered daily mean ambient temperature data (4 km grid cell resolution) for each Texas county from 1 January 2017 to 31 December 2017 obtained from the PRISM Climate Group [38]. We used EL-USB-2 Data loggers (Lascar Electronics Inc.) placed within the first third of each cave near roosting bats, when present, to record internal ambient temperature and relative humidity (RH) every hour for 1 year. Unfortunately, RH data were not reliable (% surpassed the maximum value of 100) and thus were not used. We deployed loggers at each of 27 caves (13 caves occupied by hibernating bats, 14 unoccupied) distributed in 19 counties across north and central Texas where permission was obtained. We placed loggers near bats or centrally in caves where bats were not present. We obtained information on the presence of *P. destructans* within a county from Texas Parks and Wildlife Department [39].

To use the model on Texas topography, we initially assumed all documented caves could be hibernation sites and assigned the estimated 4251 caves obtained from the Texas Speleological Survey database to 94 counties. Inside each county (a geographical region used for administrative purposes), we grouped hibernation sites according to cave mean temperatures into bins of 2°C, following a Gaussian distribution with county-specific mean and variance of 3.75°C. Each bin was considered as a patch $i$ in county $j$ in the model. The binning was done to reduce the number of patches for simulation performance reasons, and no information was lost because the locations of hibernation sites within the counties were not available to us. We estimated the mean cave temperature based on a linear model of mean ambient temperature and cave coordinates (electronic supplementary material) using the approach used in McClure *et al.* 2018 [40]. We then used this model to predict mean cave temperatures for the geographical centres of each county. We used the variance of the model residuals to estimate the 3.75°C variance.

The carrying capacity $K_{i,j}$ for patch $i$ in county $j$ was given by the number of hibernation sites aggregated in that bin (figure 1). We did not assume that all caves were occupied by bats, but rather approximated that 30% of caves (applied evenly across patches) were occupied based on 2015–2019 survey data (M.B. Meierhofer, S.J. Leivers, L.K. Wolf, K.D. Demere, J.W. Evans, J.M. Szewczak, B.L. Pierce, M.L. Morrison 2019, unpublished data). Hibernaculum temperature inside each patch varied sinusoidally with an amplitude estimated for each county (electronic supplementary material), affecting fungal growth rates inside the hibernaculum. In addition, we assigned each county a mean ambient temperature and annual sinusoidal variation, according to a linear fit (of Fourier coefficients) on the temperature data.

We implemented patch-to-patch migration (dispersal) as follows: each patch had a fixed proportion of susceptible and exposed bats emigrating per day. Given we are modelling all cave-hibernating bats as one population, we do not directly account for individual species structure variation nor movement among sites during winter. We divided the emigrating bats into recipient patches depending on the distance. We assigned a weight $w_{i \to k, j \to l} = K_{k,l} \mathrm{e}^{-\gamma d_{i \to k, j \to l}}$, where $d_{i \to k, j \to l}$ is the distance from patch $i$ in county $j$ to patch $k$ in county $l$ for each connection

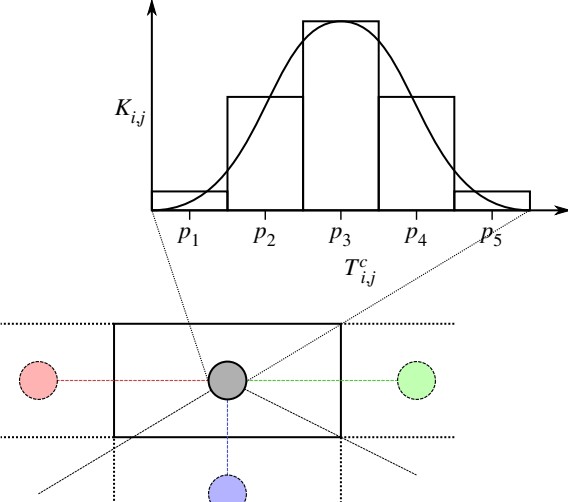

**Figure 1.** A conceptual drawing of the model spatial setting. The top part shows binning of hibernacula by the within cave mean temperatures according to a Gaussian distribution obtained from a linear model for each county $j$. Each bin becomes a patch $i$ with a given mean within cave temperature $T^c_{i,j}$ and capacity $K_{i,j}$. In the bottom part, dispersal distances between counties $j$ are the distances between the county midpoints (grey and coloured circles). (Online version in colour.)

under a cutoff distance of 100 km, and we calculated the proportion going to a target patch as $\pi_{i \to k, j \to l} = w_{i \to k, j \to l} / \left( (K_{i,j} - 1)\mathrm{e}^{-\gamma d_{j \to j}} + \sum_{n,m} w_{i \to n, j \to m} \right)$. We calculated patch to patch distances $d_{j \to j}$ within a county as the expected distances of two randomly placed points inside the county. Distances between patches in different counties were simply the distances between the two county midpoints. The parameter $\gamma$ scales the recipient patch distribution with respect to distance from the focal patch.

## (c) Model parameters and parametrization

We determined bat and fungal parameter values using referenced parameter values (averaged estimates of data sourced from literature) and approximated parameter values (averaged values of our expert opinions informed by previous survey efforts of both bats and *P. destructans* swab surveys conducted in Texas) (table 1). For infection and recovery from WNS parameters in our model, we sourced information from literature which included data from *M. lucifugus*, a species with a distribution range outside Texas. When possible, we focused on using information available on bat species known to hibernate in Texas (hibernatory bat populations) as only hibernating species are affected by WNS. The average number of non-Mexican free-tailed bats (*Tadarida brasiliensis*) was calculated to be 344 based on data collected on bat counts during our previous 2015–2019 winter survey efforts of caves in Texas (M.B. Meierhofer, S.J. Leivers, L.K. Wolf, K.D. Demere, J.W. Evans, J.M. Szewczak, B.L. Pierce, M.L. Morrison 2019, unpublished data). We disregarded *T. brasiliensis* colonies as this species does not tend to hibernate. With our previous survey counts, and documentation of large bat colonies by other researchers (e.g. [41,47,48]), we approximated that the average number of bats per cave to be 900 for the purpose of our model.

Direct assignment of parameter values to our model for the actual biological setting would have been challenging because many of the values are not known or directly measurable. Instead, we used literature-based values and approximated values of the authors in combination with a parameter estimation step based on the known initial state of the disease in 2018 and survey data from 2020. With the estimation step, we ensured

**Table 1.** The model parameters and our value estimates after the validation step based on 2020 WNS survey data. (Referenced parameter values are the averaged estimates based on data sourced from referenced publications. Approximated parameter values are the averaged values of our expert opinions informed by previous survey efforts of bats (2015–2019; M.B. Meierhofer, S.J. Leivers, L.K. Wolf, K.D. Demere, J.W. Evans, J.M. Szewczak, B.L. Pierce, M.L. Morrison 2019, unpublished data) and *P. destructans* swab surveys (2017–2019) conducted in Texas. We then scaled the aforementioned values by the carrying capacity and thus they do not directly match values provided within references.)

| symbol | parameter name | unit | value(s) | reference |
|---|---|---|---|---|
| $\hat{r}_h$ | bat population growth rate | d$^{-1}$ | 0.00333 | [41,42] |
| $r_f$ | fungal growth rate | d$^{-1}$ | 0.00152 | approximated |
| $\beta_e$ | environmental transmission rate | (unit fungi)$^{-1}$ d$^{-1}$ | 0.043 | [42] |
| $\beta_d$ | direct transmission rate | (unit bats)$^{-1}$ d$^{-1}$ | 0.195 | [19,25] |
| $\mu_h$ | hibernation mortality | d$^{-1}$ | 0.0012 | [35,43] |
| $\mu_f$ | disease mortality | d$^{-1}$ | 0.039 | approximated |
| $\lambda$ | fungal shedding | (unit bats)$^{-1}$ d$^{-1}$ | 0.017 | approximated |
| $\delta_e$ | recovery (exposed to susceptible) | d$^{-1}$ | 0.0488 | [44][a], [45] |
| $\delta_n$ | recovery (infectious to exposed) | d$^{-1}$ | 0.0225 | [44][a] [45,46] |
| $\varphi$ | infection rate | d$^{-1}$ | 0.0755 | [44][a], [45] |
| $\rho$ | migration proportion | — | 0.042 | approximated |
| $\gamma$ | migration distribution parameter | — | 0.00868 | approximated |
| init $s$ | prop. susceptible bats in initially affected counties | — | 0.7 | approximated based on swab survey results |
| init $e$ | prop. exposed bats in initially affected counties | — | 0.28 | approximated based on swab survey results |
| init $n$ | prop. infectious bats in initially infected counties | — | 0.02 | approximated based on swab survey results |
| init $f$ | free-living fungus in initially affected counties | — | 0.1 | approximated |
| $\eta^o_{h\leftarrow a,1}$ | ambient temp. threshold 1 | °C | 11.5 | [35,36] |
| $\eta^o_{h\leftarrow a,2}$ | ambient temp. threshold 2 | °C | 12.5 | [35,36] |
| $\eta^c_{h\leftarrow a}$ | hibernaculum temp. threshold | °C | 11.5 | [35,36] |
| $\hat{\omega}_{a\leftarrow h}$ | activation rate | d$^{-1}$ | 0.1 | approximated |
| $\hat{\omega}_{h\leftarrow a}$ | hibernation rate | d$^{-1}$ | 0.1 | approximated |

[a]Parameter initially estimated from the well-studied *M. lucifugus* (hibernating bat species not documented in Texas) when data for hibernating bats found in Texas were limited.

that our parameter set would predict the 2020 observed state from the initial conditions, and thus be in line with the actual known WNS disease dynamics in Texas.

To parameterize the model, we started by constructing a parameter set, which represented our best knowledge of the model parameter values obtained by averaging approximated values of the co-authors (table 1). We then refined our estimates with an approximate Bayesian computation procedure [49]. First, we constructed a prior distribution by assigning to each of the parameter values a log-Gaussian distribution with our estimate as the median value and a log-unitary standard deviation, following the reasoning that the true parameter values fall within one order of magnitude from our initial estimate. We then ran 500 simulations with parameters randomly drawn from our prior distributions (electronic supplementary material) and performed rejection sampling to select appropriate posterior combinations based on WNS 2020 survey data (figure 2). With no easy way of assigning likelihood values to our simulations, we used simple rejection thresholds. We used $\theta_d < 0.1\%$ disease prevalence in counties where WNS in bats was detected as the rejection criteria, following the reasoning that a small prevalence in bats could already be detected through surveys. We further used $\theta_f < 20\%$ free-living fungus prevalence in counties where *P. destructans* was detected as the rejection criteria, because

finding fungal growth outside of the bat hosts requires an active search of hibernation sites after 2 years of simulation time. Our threshold values were admittedly arbitrary because we had no information on the actual detection effort or efficiency, but these values can easily be improved in future work. Additionally, there were two counties surveyed with neither WNS or *P. destructans* detected, and we rejected greater than 0.1% disease prevalence and greater than 20% free-living fungus prevalence in these. We then used parameter median values from the accepted combinations (with 5% acceptance rate) as our validated parameter set. We fixed hibernation rate and threshold parameters to our literature-based estimates. We studied the robustness of our results separately in a sensitivity analysis (electronic supplementary material), where we investigated how varying each parameter by a small increment or decrement changes the simulation outcome in terms of the number of affected patches and reduction in bat numbers.

## (d) Analysis of model outcomes

We visualized model predictions in *R* with interpolated heat maps generated by the linear bivariate method in the package 'akima::interp()' on an 80 × 80 grid under default settings. Interpolation predicts values within a convex hull bounding the data

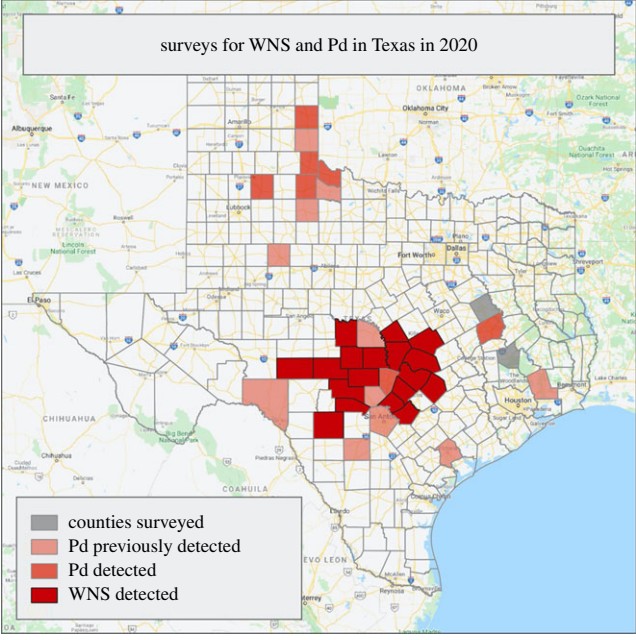

**Figure 2.** Texas counties where WNS was detected in 2020 (dark red), where only *P. destructans* was detected in 2020 (medium red), where *P. destructans* was detected in previous years (light red), and counties surveyed in Texas where neither WNS nor *P. destructans* was detected (grey). (Online version in colour.)

points. Therefore, we did not predict beyond the spatial extremes of the data produced by the predictive model described above. We plotted infection as the carrying-capacity-scaled predictions from the infection model at 5 and 10 years of simulation and calculated the loss of bat abundance as the proportional reduction in bats predicted by the infection model relative to a no-infection scenario, obtained by running the model without the fungus and initial infections. We used functions in the 'Raster' and 'ggplot2' packages to create the figures.

## 3. Results

In total, there were 4251 hibernation sites of 14 132 potential sites occupied by bats aggregated into 293 patches within our model. Under our parameter set validated against 2020 WNS survey data, the bat population declined 35.6% across 84 counties in 10 years (figure 3). After 5 years, we found the bat population will be reduced by 19.3% in 70 counties. The simulations did not show local extinctions in any county, but the bat population reduced by 86% (85% after 5 years) in the most affected site. The most affected counties were in north Texas, with *P. destructans* present at the start of the simulation. The bat population rich mid-Texas counties are projected to lose between one quarter to half of the bat population (figures 3*c*,*d* and 4*a*). The density of the fungus and its spores reached high levels in these counties (figure 4*b*).

*Pseudogymnoascus destructans* caused low mortality in the southernmost counties under our parameter set because high ambient temperatures did not support long enough hibernation periods for significant disease progression to the infectious state (figure 4*a*). The warm temperatures and resulting short hibernation period also reduced the impact in central Texas. While cold patches may have periods of hibernation even in warm counties, the cave temperature was then below optimal (13.0°C, [22]) for fungal growth.

Exposed bats carrying the fungus will be present, however, because of dispersal from affected sites.

While both transmission modes—environmental and direct—are significant components of epidemic spread, under our parameterization, transmission via the environment had a larger impact, causing approximately 90% of the force of infection along the simulation time (electronic supplementary material). However, sensitivity analysis on the infectivity parameters shows that similar results can be obtained by decreasing one parameter and increasing the other parameter (i.e. adjusting rate parameters associated with the two transmission modes; electronic supplementary material). Removing environmental transmission from the model resulted in 99% less bat population reduction after 10 years. This occurs because most of the exposed and infected bats shed the fungus and revert back to the susceptible state during the summer, and transmissions from the environment is required to re-infect the bat population at the start of the hibernation period.

The sensitivity analysis shows that the spread of WNS changes under variation of the parameters. Specifically, our results are most susceptible to changes in direct transmission rate, infection rate, hibernation temperature thresholds and bat growth rate. The increase in the direct transmission rate, infection rate and hibernation temperature thresholds increase the disease mortality and fungal spread. An increase in bat population growth rate decreases mortality and spread. Increasing the mean dispersal distance (decreasing $\gamma$) significantly increases the number of affected patches, but does not significantly affect mortality.

Under sigmoidal infectivity response, the range of potential outcomes is wider (electronic supplementary material). Depending on the parameterization of the sigmoid curve, we could expect the number of affected patches in 10 years to range from 30 to 70, with population reductions ranging from 8 to 40%. Unfortunately, to our knowledge, there are no experimental data available that would allow inferring the true shape of the infectivity response.

## 4. Discussion

We found that WNS mortality will vary across Texas cave hibernacula, with northern sites more affected than southern sites. Results from our model suggest a projected decline (greater than 75% reduction) in the number and size of bat populations in the northern sites over 5–10 years. Central sites will be affected to a lesser degree, with a projected 30–50% reduction in population densities, whereas southern sites will be mostly unaffected. Interestingly, *P. destructans* reaches very high densities in central Texas where hibernation sites are most numerous, but owing to warm temperatures, the bat populations in these sites are less severely affected probably owing to shorter periods of time spent in torpor. The high fungal densities are in part explained by the reduced mortality, resulting in a long duration during which bats shed fungal spores.

The first documentation of WNS was anticipated to be in north Texas based on environmental characteristics [35] and proximity to nearest WNS-infected sites. However, WNS was first documented on cave myotis (*M. velifer*) in 18 counties in central Texas in spring 2020 ([34]; figure 2). This is the first documentation of WNS in central and southern regions, resulting after 4 years of *P. destructans* being present in Texas. Our

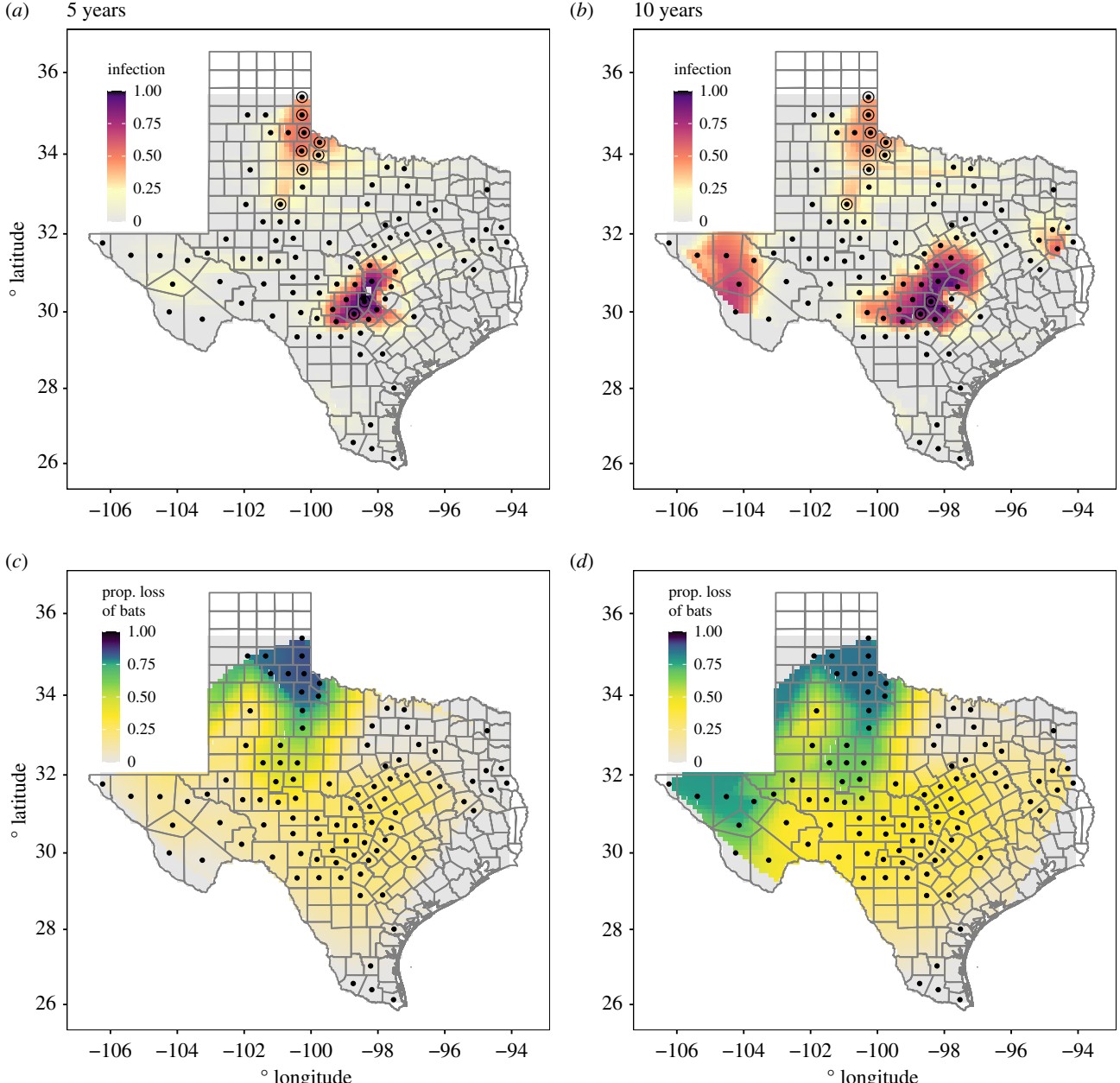

**Figure 3.** Interpolation of the carrying-capacity-scaled infection output for (*a*) 5 and (*b*) 10 years of simulation. Grey tiles show regions of 0 predicted infection. Interpolation predicts for values within a convex hull around the county centre points (black dots). Sites with observed infection data in 2018 marked with a circle around the point. Interpolation of the proportional loss of bats relative to infection-free model for (*c*) 5 and (*d*) 10 years. Gradient scale of heat map weighted to distinguish between larger degrees of loss (50%+). (Online version in colour.)

model shows that both the fungus and WNS are prevalent in central Texas, but that the proportional disease mortality is smaller in central Texas than in northern Texas. Because few exposed bats die in central Texas, *P. destructans* can reach high densities, increasing the continued spread of the fungus. Central Texas has the greatest abundance of known hibernacula in Texas, as well as the greatest diversity of bats in the state [47], increasing the potential for infection susceptibility. Unfortunately, bats found with WNS in central Texas during early spring were found by the general public outside of their hibernacula, and it is unknown where bats are becoming infected with *P. destructans* in the region.

Based on our model, the environmental transmission may play an important part in the spread of the epidemic. Indeed, contact between bats and the contaminated environment [50] in autumn has been shown to initiate infection [51]. This also complements the recent finding that high levels *P. destructans* in the environment result in widespread infections [52]. Although the primary method of spread of *P. destructans* is bat-to-bat [13], under our parameter set only 1 in 10 is owing to direct contact with an infectious individual during hibernation. The overall pattern is not very sensitive to the relative strengths of these two components (modes of pathogen spread: environmental, direct) and temporally detailed data would be required to estimate these parameters independently. This is important to note, however, as indirect and infrequent transmission plays a key role in the transmission and community-wide spread of *P. destructans* [53]. Indeed, direct transmission still impacted the bat population; removing direct transmission resulted in 90% less bat population reduction. Direct transmission is the most probable cause of pathogen spread into new counties. When exposed

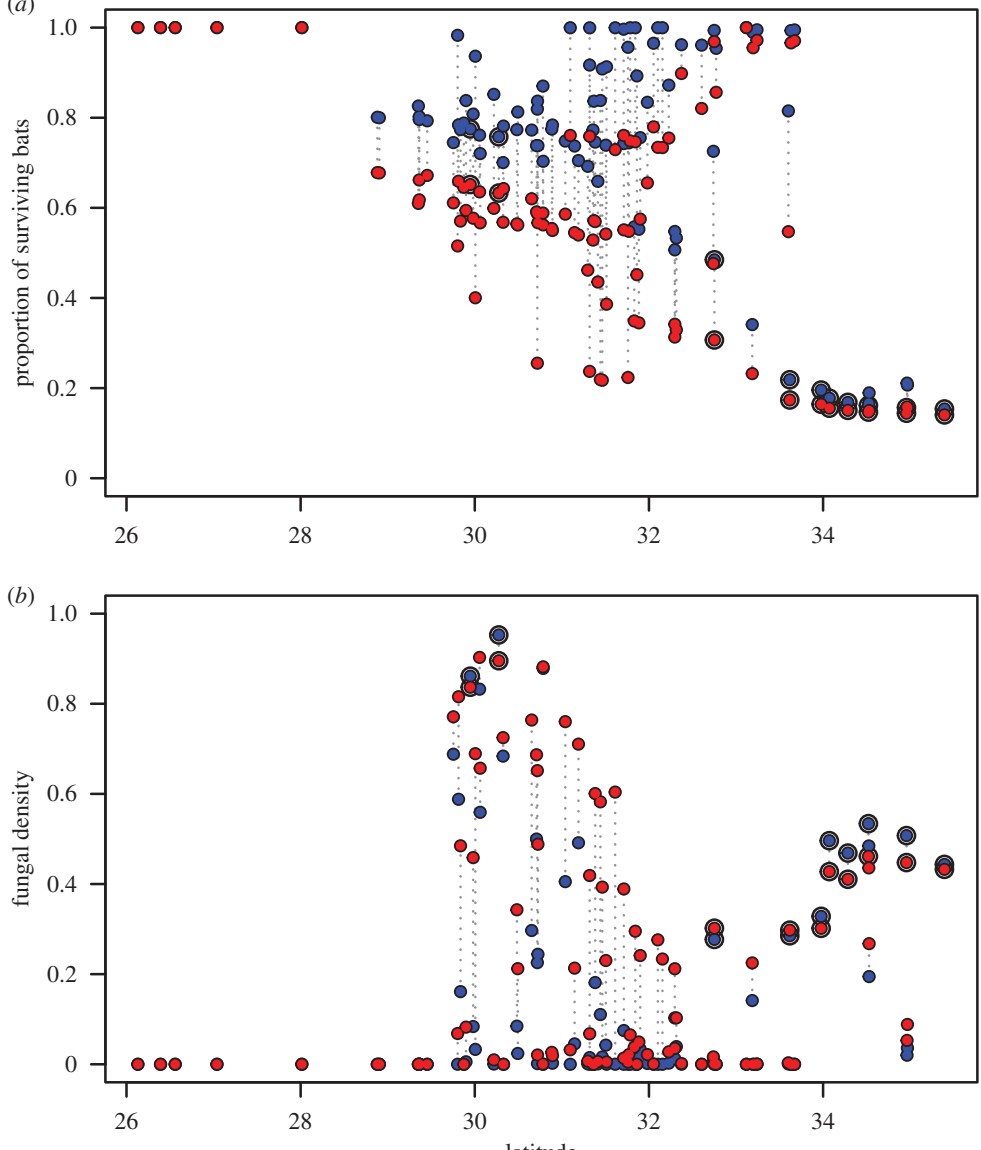

**Figure 4.** Effects of the disease on counties at different latitudes. The proportion of surviving bats (*a*) and the amount of free-living fungus (*b*), averaged for each county. The effects after 5 and 10 years are marked with blue and red dots, respectively. The matching counties are connected by dotted lines. Counties initiated with the disease are marked with a black circle around the dot. The *y*-axis scale on (*b*) is in units of fungal carrying capacity. (Online version in colour.)

bats disperse into new sites, they shed fungus into the environment, but also transmit directly to susceptible hosts when entering hibernation. Because fungal densities remain low in the environment at new sites initially, the direct transmission route may be more prevalent.

Our results suggest that reducing fungal spore loads in hibernation sites may work as an effective way to slow down the epidemic spread. Susceptibility to the disease requires bats to stay in torpor for prolonged periods, suggesting that pathological infection occurs in regions with long periods of low ambient temperature [54]. Indeed, knowledge of hibernation temperatures of several species in Texas [35] supports the notion of longer periods of the torpor of bats in north Texas than in central and southern regions of Texas. Further, the known largest bat colonies in the world exist in Texas [47], with some colonies of hibernating bat species occurring statewide in the thousands (e.g. *P. subflavus*, [55]) to tens of thousands (e.g. *M. velifer*, [48]). These large colonies can provide environments conducive to the persistence of organic detritus, supporting vegetative growth of *P. destructans* and creating sources of increased

potential environmental transmission [42]. RH is also known to constrain propagation of *P. destructans* [23]. Unfortunately, we did not include RH in the model because sufficient data were not available. However, we know that bats also tend to hibernate at sites with high RH, to reduce evaporative water loss [37]. Conidial fungi, such as *P. destructans*, also need high RH to propagate [56] and grow [23]. Essentially, RH affects the hibernation success of bats independently of the presence of *P. destructans* [57,58].

The projections are dependent on our parameterization of the dynamical model. Finding the relevant parameter set for a particular case is admittedly difficult, despite that for some parameters the values were available from previous work [27]. Further, some parameters such as transmission rate are dynamical and therefore can only be estimated. We performed an additional validation step to refine our estimates based on observations of disease spread two years since introduction to Texas. Our initial best-estimate parameter values undershot the observed pattern of disease prevalence in 2020; refined estimates after validation led to the faster spread and less lethal disease (electronic supplementary

material). The values we chose for rejection thresholds (rejection criteria) also reflect our subjective views of at which level of prevalence the WNS disease and *P. destructans* would be detected in surveys. Similar analysis and model projections could be performed with our model framework in the following years when new data become available, thereby improving estimates and predictions.

We anticipate that the spread of *P. destructans* will be slow and display source-sink dynamics in Texas. We further anticipate that the spread of *P. destructans* in central Texas, where caves are more clustered, will be similar to the eastern United States, where the rate of spread increased with proximity to the nearest infected site [20]. Indeed, results from our model suggest that conservation actions should consider preservation of sites in north Texas that have temperatures conducive to hibernation and suitable for fungal growth, with large colonies of bats, as these sites may be more susceptible to local extinctions [25]. Further efforts should be focused on gathering species-specific parameters and within-season movements at southern latitudes for furthered targeting efforts for future research.

**Ethics.** We followed field hygiene protocols in accordance with United State Fish and Wildlife Service WNS Decontamination Guidelines. We obtained written permission for cave access from the appropriate landowners and managers.

**Data accessibility.** The datasets supporting this article have been uploaded as part of the electronic supplementary material.

**Authors' contributions.** M.M.: conceptualization, data curation, methodology, project administration, writing—original draft, writing—review and editing; T.L.: conceptualization, investigation, resources, supervision, writing—original draft, writing—review and editing; L.R.: conceptualization, formal analysis, methodology, validation, visualization, writing—review and editing; J.S.J.: data curation, investigation, writing—original draft, writing—review and editing; S.P.: investigation, methodology, visualization, writing—review and editing; M.L.M.: conceptualization, funding acquisition, project administration, resources, supervision, writing—review and editing; B.L.P.: conceptualization, funding acquisition, project administration, resources, supervision, writing—review and editing; J.W.E.: conceptualization, funding acquisition, writing—review and editing; J.A.: conceptualization, data curation, formal analysis, methodology, validation, visualization, writing—original draft, writing—review and editing. All authors gave final approval for publication and agreed to be held accountable for the work performed therein.

**Competing interests.** We declare we have no competing interests.

**Funding.** Funding for this project was provided through the U.S. Fish and Wildlife Service's State Wildlife Grant Program (CFDA no. 15.611) as administered by Texas Parks and Wildlife Department and the U.S. Fish and Wildlife Service (CFDA no. 15.657). Additional funding was provided by the fight WNS 'Micro Grants for Microbats' and the NSS WNS Rapid Response Fund. This work was supported by the Fulbright Finland Foundation and Finnish National Agency for Education (EDUFI) and the Academy of Finland (grant no. 331515).

**Acknowledgements.** D. Wright, Dr Comer and Dr Godwin are thanked for site access; J. Kennedy is thanked for site access and data collection; J. Carey, K. Demere, S. Goree, L. Johnston, B. Kimbell, S. Leivers, B. Stamps, E. Whittle and L. Wolf are thanked for field assistance. We thank the journal referees for their insightful and helpful review comments.

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
