## [Peer Review File · Proceedings of the Royal Society B: Biological Sciences]

Review History

RSPB-2020-2901.R0 (Original submission)

Review form: Reviewer 1

Recommendation

Major revision is needed (please make suggestions in comments)

Scientific importance: Is the manuscript an original and important contribution to its field?

Good

General interest: Is the paper of sufficient general interest?

Acceptable

Quality of the paper: Is the overall quality of the paper suitable?

Acceptable

Is the length of the paper justified?

Yes

Should the paper be seen by a specialist statistical reviewer?

No

Do you have any concerns about statistical analyses in this paper? If so, please specify them explicitly in your report.

No

It is a condition of publication that authors make their supporting data, code and materials available - either as supplementary material or hosted in an external repository. Please rate, if applicable, the supporting data on the following criteria.

Is it accessible?

Yes

Is it clear?

Yes

Is it adequate?

Yes

Do you have any ethical concerns with this paper?

No

Comments to the Author

The paper "Ten-year projection of white-nose syndrome disease dynamics at the southern leading-edge of infection in North America" is a well thought-out manuscript that fits the scope of the journal and meets the journal's standards of originality, quality and language. The authors present a means of predicting white-nose syndrome spread based on cave densities and average dispersal distances of bat species, in addition to predicting the impacts of WNS on bat populations in Texas. Though potential impacts of WNS on populations has been modeled across the United States, specifically taking into account the spatial variation in environmental conditions that impact the causal fungus is timely. The work is interesting and will add a lot of important knowledge to management of this devastating disease. There were some spots of the manuscript that require clarification, but I don't want these issues to take away from the value of this work nor the appropriateness of the modeling approach.

My one main concern is of the complexity of the modeling effort and the ability for this approach to be described well enough for the reader to understand. I applaud the authors for trying to tackle these source-sink disease dynamics in a cave system with a hard-to-track animal, but I do think there needs to be some more clarity in how the model is presented, applied, and parameterized. I was left confused in how all of the many pieces fit together and how the many parameters were defined. I actually was a peer reviewer on the Lilley et al. (2018) manuscript, and I recall having a hard time wrapping my head around the modeling approach. Therefore I am unable to comment on the validity of the results presented here, as I was unsure how they were obtained. Additionally, there should be some careful consideration in how these results are discussed in the discussion: these are predictions, based on simplifications of a system using parameters averaged over multiple species and locations. Therefore the authors should take care in making broad statements in the discussion given these results. Instead the authors should note the predicted declines rather than taking these results as hard truths.

I understand why the methods were set up in terms of describing the model first and then the parameterization, but it makes it difficult to go back and forth between how the parameters are used and how they were measured. I suggest reorganizing this entire section into sub-sections based on each model component, where the model component can be described and then discussed where the data come from the parameterization of that component.

Below I outline the line-specific comments that could hopefully clarify how the model is applied in this system.

Line by line comments:

Key words: write out white-nose syndrome; may be also appropriate to include *Pseudogymnoascus destructans*.

Line 48: Since there are so many other abbreviations in the manuscript, and EID is only referenced 3 times, it would be more appropriate to not use the abbreviation.

Line 56: And host physiology, such as the case with WNS.

Line 58: and behavior

Line 63: mortality of the host

Line 78: Pet peeve: though WNS itself is not what is being spread, but rather the fungus which then causes the disease. Change “WNS disease dynamics spread” to “fungal spread”

Line 86: Break up into two separate sentences instead of having the comma between the United States and many

Line 88: again, WNS is not invading, the fungus is. Change “infection” to “observation”

Line 103: clarify from what the demographics and environmental data referenced here are measured from (i.e. bat demographics and hibernacula environmental data)

Line 109: Please clarify this hypothesis: by stating “high cave concentrations” do you mean high concentrations of caves in an area? High concentrations of bats in a cave? And by stating “bat abundance,” do you mean bat abundance across the landscape? Bat abundance within the cave (and therefore density)? And on this thought, do you consider how cave size can therefore affect dynamics regarding bat abundance – the same number of bats within a small cave vs. a larger cave should impact fungal spread.

Line 111: Should ecology really be physiology here?

Line 111: It would be helpful to see some predictions following the hypotheses. This would allow the reader to know exactly what the paper is testing to confirm or falsify these hypotheses.

Line 123-126: Since the model is not described within the text, and the descriptions in the Supp Materials are mathematically dense, it may be worthwhile to include a flow chart that shows the different components of the model (such as in Figure 1 in Lilley et al. 2018). Specifically, what is modified from the Lilley et al. (2018) model that is then applied in this manuscript?

Line 131: thank you for including your GitHub repository!

Line 135: TSS not needed

Line 137: Why were the sites grouped into temperature bins? How is that applied later on? It is not clear here.

Line 140: thank you for including your code.

Line 143: I see now that the temperature bins are the “patches” – but these are not “patches” in a geographic sense, correct? Would distance between caves therefore influence spread?

Line 145-146: How was this 40% applied across patches? And what survey data?

Line 146: Patch number should be moved to results

Lines 154-157: Could this be represented in a figure to visualize these parameters?

Lines 269-273: this sounds like discussion rather than just reporting results

Table 1: It is unclear if each of the “approximated” parameters were approximated in this manuscript or elsewhere. Perhaps either add more detail to the approximation method in the table, or outline the methods section to match the different parameters in the table to the reader can reference the text easily.

Figure 1: This figure is hard to interpret. Either the figure needs to be clearer in its presentation and/or the legend needs to be more specific in how to interpret.

Review form: Reviewer 2

Recommendation

Accept with minor revision (please list in comments)

Scientific importance: Is the manuscript an original and important contribution to its field?

Excellent

General interest: Is the paper of sufficient general interest?

Excellent

Quality of the paper: Is the overall quality of the paper suitable?

Excellent

Is the length of the paper justified?

Yes

Should the paper be seen by a specialist statistical reviewer?

Yes

Do you have any concerns about statistical analyses in this paper? If so, please specify them explicitly in your report.

Yes

It is a condition of publication that authors make their supporting data, code and materials available - either as supplementary material or hosted in an external repository. Please rate, if applicable, the supporting data on the following criteria.

Is it accessible?

Yes

Is it clear?

Yes

Is it adequate?

Yes

Do you have any ethical concerns with this paper?

No

Comments to the Author

Please see attached file. (See Appendix A)

Review form: Reviewer 3

Recommendation

Reject - article is scientifically unsound

Scientific importance: Is the manuscript an original and important contribution to its field?

Marginal

General interest: Is the paper of sufficient general interest?

Marginal

Quality of the paper: Is the overall quality of the paper suitable?

Marginal

Is the length of the paper justified?

Yes

Should the paper be seen by a specialist statistical reviewer?

Yes

Do you have any concerns about statistical analyses in this paper? If so, please specify them explicitly in your report.

No

It is a condition of publication that authors make their supporting data, code and materials available - either as supplementary material or hosted in an external repository. Please rate, if applicable, the supporting data on the following criteria.

Is it accessible?

No

Is it clear?

No

Is it adequate?

No

Do you have any ethical concerns with this paper?

No

Comments to the Author

I think the authors did an exceptional job working with the data and models available to make 5 and 10-year projections of Pd spread and WNS mortality for Texas. Clearly, models like this are heavily dependent on initial conditions and underlying assumptions. So the question becomes, are the assumptions and data on initial conditions sufficient to support the results and conclusions? In particular my main concerns are, infectious parameters appear drawn from a species not found in Texas (MYLU), all hibernating bat species are modeled as one population – when we know there are inter-specific responses to Pd and striking inter-specific differences of WNS mortality among hibernating bats, and several parameters were derived from a validation exercise that had no data on surveillance/detection effort. The methods section is currently poorly written and jumps around. Careful revisions to this section may help alleviate these concerns. More explicit recognition of which parameters are pulled from MYLU and how these may or may not be applicable to an amalgam of all Texas hibernating bats is also needed. Overall these issues leave me with deep concerns about the validity of the results and conclusions. I'm keen on the premise, I'm just not convinced the data available are sufficient for the task at hand.

The authors made a choice, due I suspect to lack of species-level data, to model all hibernating bats as one population, which ignores known inter-specific threats from WNS. It also fails to mention that only one species, MYVE appears to actually get WNS in Texas (as of 2020), though I think the community is waiting to see what happens to PESU. Also unclear are which 'species' contributed parameter values that were then lumped together in this analysis, in a sort of hypothetical Frankenstein 'Texas hibernating bat'. I would recommend the authors take a more conservative approach and focus their attention on MYVE and PESU alone, and carefully point out which infection parameters are derived from MYLU and disease ecology in the NE, and to the extent possible, remove this source of unnecessary bias. If species level data are not available it's understandable why the authors went with the route they did, so when MYLU parameters are used they should be contextualized for the reader as MYLU are not known to typically range in Texas. For example, how would differences in water vapor deficit (cave environment) and evaporative water loss (host) impact projections on fungal growth/shedding between the NE and Texas? The author's method section was challenging to understand. I think it needs to first focus

on carefully explaining the measured data and then describe how these data were modeled. The current organization jumps around and leads to unnecessary and avoidable reader confusion. I also was concerned that the detection effort in 2018 and 2020 is reported as unknown, which will introduce considerable bias into the parameterization 'validation step'.

53 - 'new habitable environmental reservoir pathogens' - is unclear. Perhaps you mean something along the lines of 'new habitat for pathogens found in environmental reservoirs'?

77 - would 'population demographics' be more appropriate than 'population composition'?

89-92 - This is a long sentence that is a lot to take in for a reader. The last 2 chunks of 'minimal information' are presented without citations as to their importance, and their importance could use some further explanation. I.e. help the reader understand why you think the spatial layout of caves, which could mean either the spatial distribution of caves on the landscape or the spatial arrangement of microclimates and bats within a cave, relevant? Also why is the frequency of suitable caves relevant? As you point out in the next sentence, Pd is free living and can exist in environments absent bats.

90-91 - why the focus on caves here when certainly other hibernacula are present (culverts, buildings)? (confirmed below in the methods section)

107 - If this is the case, the Abstract is misleading at line 32, as the model is not species-specific.

123 - Concerningly, the Lilley et al. 2018 model was parameterized for MYLU (which are not found in Texas) and for WNS occurrence conditions in the Northeast U.S.. Methods provide no indication that host-associated parameters were adapted for Texan hibernating bat species (i.e. direct transmission rate, infection rate, carrying capacity, recovery rates, population growth rate) nor the appropriateness of applying this model to describe WNS threats in Texas. However, in line 199 it becomes clear the model was somewhat adapted to hibernating bats in Texas. A careful reorganization of the method section would provide more clarity to the reader.

135 - Unclear - you had environmental data from 94 counties or for 4251 caves? Thereafter unclear how measured environmental data from caves is used as the McClure et al. 2020 model is described as being used to predict mean cave temps for the geographic centers of each county. More confusion is added once I reached line 191 where collection of cave temperature data is described. Even more confusion is added in Supplement 1 where equations used to calculate ambient conditions are provided (S1 and S2). Perhaps this section of the manuscript can be reorganized so the reader is introduced first to the data collected and then how the model uses it.

137 - What is the temporal and spatial resolution of the temperature data used to calculate cave mean temp?

140 - Should clarify if the reference is for the approach used in McClure et al. 2020 or if the model from McClure et al. 2020 is being used

146 - Unclear how total number of patches were arrived at with binning etc.

150 - PRISM data needs a citation

153 - Not accounting for species structure seems questionable when WNS is only documented in MYVE in Texas along with others that are positive for Pd by not WNS.

166 - Notation is difficult to read, looks like it needs some extra spacing -

167 - Any data to support MYVE uses torpor at temps up to 12.5°C?

175 - The sigmoidal infectivity does have notable effects on disease dynamics - as shown in the Supplement 1. Given their huge effects (at 10 years the sigmoid-effect estimated there would be ~30 affected counties compared to 70 to 80 counties with the linear-effect) it's worthy of another mention in the results to give a range of potential outcomes.

201 - The parameter values came from which species specifically? When parameters were not available for Texas bat species, did you default back to parameters for MYLU?

202 - Only one hibernating bat species is reported with WNS in Texas, MYVE, so why not just model this species and PESU for a more conservative approach (and any other confirmed WNS species found in Texas I'm overlooking). This will have implications on the bat mortality estimates - which may likely be overestimates.

211 - How extensively was Texas surveyed in 2018 and 2020? If the survey in 2018 was partial would that introduce bias to the approximated values used in the 5-19 year modeling steps? This is further concerning as line 230 indicates the authors had no information on the actual detection effort or efficiency.

236 - Something unclear here - 'and were not found by this validation step'?

247 - needs 'counties' after mid Texas?

249 - 'rich' used again - repetitive to reader

274 - Very concerning that the model is most sensitive to changes in direct transmission rate and infection rate - that are pulled from studies of different bat species especially as we know there are interspecific differences in these parameters. Not mentioned here - importantly it also appears sensitive to amount of free-living fungus in initially affected counties - which was approximated and speaks to my concern about detection effort and bias. It was also sensitive to bat pop growth rate (which was based on MYLU).

275 - Much too long of a sentence for readers.

281 - The contributions of humidity to WNS susceptibility/outcomes is overlooked and should be brought into the discussion to further contextualize the discussion of temperature results.

296 - The finding of WNS in central Texas in the model is not at all surprising because it was parameterized using data from 2018-2020 and WNS was already detected in central Texas in 2020 - so the model is not actually making a new prediction about where WNS will be found in Texas.

336 - Needs grammatical work on the comparative if you want to use 'than' i.e. longer periods of torpor...

344 - Agree completely - including species specific variation!

348 - The model is focused on Texas so I would temper this statement as geographies outside of Texas may also contribute to the source-sink dynamics. Also how does this square with the early circa 2006-2010 Pd spread basically occurring from north to south (opposite of the direction you propose)? Does the lack of karst habitat across vast sections of the mid-west speak to the anticipated dynamics?

358 - Though again these are for MYLU

360 - Does 2 years of pathogen spread with unknown detection effort really provide a solid baseline for parameterizing this type of model?

374 – Source-sink dynamics are more about where the pathogen spreads from and to, it doesn't necessarily speak to susceptibility and potential for local extinctions, which is more about temp/humidity conditions and at what temperatures different species use torpor.

379 – Why is movement data the best way to estimate infectiousness? Please expand. I'm wondering about this conclusion as I interpret infectiousness here to be concerned with understanding aspects of direct transmission. Given the finding that 90% of the force of infection comes from environmental transmission, environmental measures of temp/humidity will likely give you the best indication of Pd suitability, and the observation that Pd is already found in the north and central regions of Texas how will this knowledge improve management decisions. To add to the last argument, the model is also predicting pretty much every population will have fungal spores in 5-10 years (Appendix S10). Also surprisingly no mention of looking at measuring species-specific parameters for further targeting efforts for future work?

576 – Mathematical notation issues in the creation of the pdf I suspect

587 – I think you mean 2018?

Supplement 1

- Stick/check with British or American spellings throughout (parameterization vs. parameterisation)
- Please address '(ref Anttila)'
- Standardize use of your Pd abbreviation throughout (i.e. not pd)
- 'initially affected patches (2017)' – wasn't it based on 2018 data?
- Figure titles shouldn't include variable symbols that require a new reader to visit a table to translate, rather use the parameter name itself.
- S8-S9 – consider revisiting your color gradient so it is more informative for this figure (everything looks vaguely yellow)
- S10 – is a helpful figure for readers – consider bringing this into the main narrative as a figure!

Supplement 2

2 Data – provide resolution spatial and temporal of the PRISM data you use

2.3 How was the filter for 'winter' duration, i.e. hibernation period determined? Perhaps provide a reference from roost acoustic activity?

Generally confused as the main text references McClure et al. 2020 as the source of the model for temperature data for caves – but after reviewing the code, it seems a fix might be to clarify the reference in the main text is using the approach used in McClure et al. 2020 and not that specific model.

Needs a spellcheck – see 'therefor'

Appendix S7

Can the authors provide some context on what range of sigmoid crossing point values are applicable?

Decision letter (RSPB-2020-2901.R0)

02-Feb-2021

Dear Dr Meierhofer:

I am writing to inform you that your manuscript RSPB-2020-2901 entitled "Ten-year projection of white-nose syndrome disease dynamics at the southern leading-edge of infection in North America" has, in its current form, been rejected for publication in Proceedings B.

This action has been taken on the advice of referees, who have recommended that substantial revisions are necessary. With this in mind we would be happy to consider a resubmission, provided the comments of the referees are fully addressed. However please note that this is not a provisional acceptance.

Sincerely,
 Professor Hans Heesterbeek
 mailto: proceedingsb@royalsociety.org

Associate Editor
 Board Member: 1
 Comments to Author:

This manuscript tackles an important topic and the referees uniformly support the efforts and approach.

However, they make detailed and critical comments especially regarding the choices around model and parameterization. In developing the next version of the manuscript, I think it is critical that the authors address:

- (i) clarity, so that readers can follow what is being done with the model to generate the results;
- (ii) justification of model assumptions (especially the one-population approximation - I think that details on how multiple species interactions might affect this should be given) and how parameters were selected/estimated (again, acknowledging that species/locations may have different parameters);
- (iii) furthermore, since there are parameters here that are hard to estimate or questionable, performing a global sensitivity analysis would partially allay concerns around the sensitivity of the model to these assumptions/estimates.

Reviewer(s)' Comments to Author:

Referee: 1

Comments to the Author(s)

The paper “Ten-year projection of white-nose syndrome disease dynamics at the southern leading-edge of infection in North America” is a well thought-out manuscript that fits the scope of the journal and meets the journal's standards of originality, quality and language. The authors present a means of predicting white-nose syndrome spread based on cave densities and average dispersal distances of bat species, in addition to predicting the impacts of WNS on bat populations in Texas. Though potential impacts of WNS on populations has been modeled across the United States, specifically taking into account the spatial variation in environmental conditions that impact the causal fungus is timely. The work is interesting and will add a lot of important knowledge to management of this devastating disease. There were some spots of the manuscript that require clarification, but I don't want these issues to take away from the value of this work nor the appropriateness of the modeling approach.

My one main concern is of the complexity of the modeling effort and the ability for this approach to be described well enough for the reader to understand. I applaud the authors for trying to tackle these source-sink disease dynamics in a cave system with a hard-to-track animal, but I do think there needs to be some more clarity in how the model is presented, applied, and parameterized. I was left confused in how all of the many pieces fit together and how the many parameters were defined. I actually was a peer reviewer on the Lilley et al. (2018) manuscript, and I recall having a hard time wrapping my head around the modeling approach. Therefore I am unable to comment on the validity of the results presented here, as I was unsure how they were obtained. Additionally, there should be some careful consideration in how these results are discussed in the discussion: these are predictions, based on simplifications of a system using parameters averaged over multiple species and locations. Therefore the authors should take care in making broad statements in the discussion given these results. Instead the authors should note the predicted declines rather than taking these results as hard truths.

I understand why the methods were set up in terms of describing the model first and then the parameterization, but it makes it difficult to go back and forth between how the parameters are used and how they were measured. I suggest reorganizing this entire section into sub-sections based on each model component, where the model component can be described and then discussed where the data come from the parameterization of that component.

Below I outline the line-specific comments that could hopefully clarify how the model is applied in this system.

Line by line comments:

Key words: write out white-nose syndrome; may be also appropriate to include *Pseudogymnoascus destructans*.

Line 48: Since there are so many other abbreviations in the manuscript, and EID is only referenced 3 times, it would be more appropriate to not use the abbreviation.

Line 56: And host physiology, such as the case with WNS.

Line 58: and behavior

Line 63: mortality of the host

Line 78: Pet peeve: though WNS itself is not what is being spread, but rather the fungus which then causes the disease. Change “WNS disease dynamics spread” to “fungal spread”

Line 86: Break up into two separate sentences instead of having the comma between the United States and many

Line 88: again, WNS is not invading, the fungus is. Change “infection” to “observation”

Line 103: clarify from what the demographics and environmental data referenced here are measured from (i.e. bat demographics and hibernacula environmental data)

Line 109: Please clarify this hypothesis: by stating “high cave concentrations” do you mean high concentrations of caves in an area? High concentrations of bats in a cave? And by stating “bat

abundance,” do you mean bat abundance across the landscape? Bat abundance within the cave (and therefore density)? And on this thought, do you consider how cave size can therefore affect dynamics regarding bat abundance – the same number of bats within a small cave vs. a larger cave should impact fungal spread.

Line 111: Should ecology really be physiology here?

Line 111: It would be helpful to see some predictions following the hypotheses. This would allow the reader to know exactly what the paper is testing to confirm or falsify these hypotheses.

Line 123-126: Since the model is not described within the text, and the descriptions in the Supp Materials are mathematically dense, it may be worthwhile to include a flow chart that shows the different components of the model (such as in Figure 1 in Lilley et al. 2018). Specifically, what is modified from the Lilley et al. (2018) model that is then applied in this manuscript?

Line 131: thank you for including your GitHub repository!

Line 135: TSS not needed

Line 137: Why were the sites grouped into temperature bins? How is that applied later on? It is not clear here.

Line 140: thank you for including your code.

Line 143: I see now that the temperature bins are the “patches” – but these are not “patches” in a geographic sense, correct? Would distance between caves therefore influence spread?

Line 145-146: How was this 40% applied across patches? And what survey data?

Line 146: Patch number should be moved to results

Lines 154-157: Could this be represented in a figure to visualize these parameters?

Lines 269-273: this sounds like discussion rather than just reporting results

Table 1: It is unclear if each of the “approximated” parameters were approximated in this manuscript or elsewhere. Perhaps either add more detail to the approximation method in the table, or outline the methods section to match the different parameters in the table to the reader can reference the text easily.

Figure 1: This figure is hard to interpret. Either the figure needs to be clearer in its presentation and/or the legend needs to be more specific in how to interpret.

Referee: 2

Comments to the Author(s)

Please see attached file.

Referee: 3

Comments to the Author(s)

I think the authors did an exceptional job working with the data and models available to make 5 and 10-year projections of Pd spread and WNS mortality for Texas. Clearly, models like this are heavily dependent on initial conditions and underlying assumptions. So the question becomes, are the assumptions and data on initial conditions sufficient to support the results and conclusions? In particular my main concerns are, infectious parameters appear drawn from a species not found in Texas (MYLU), all hibernating bat species are modeled as one population – when we know there are inter-specific responses to Pd and striking inter-specific differences of WNS mortality among hibernating bats, and several parameters were derived from a validation exercise that had no data on surveillance/detection effort. The methods section is currently poorly written and jumps around. Careful revisions to this section may help alleviate these concerns. More explicit recognition of which parameters are pulled from MYLU and how these may or may not be applicable to an amalgam of all Texas hibernating bats is also needed. Overall these issues leave me with deep concerns about the validity of the results and conclusions. I’m keen on the premise, I’m just not convinced the data available are sufficient for the task at hand.

The authors made a choice, due I suspect to lack of species-level data, to model all hibernating bats as one population, which ignores known inter-specific threats from WNS. It also fails to mention that only one species, MYVE appears to actually get WNS in Texas (as of 2020), though I think the community is waiting to see what happens to PESU. Also unclear are which ‘species’

contributed parameter values that were then lumped together in this analysis, in a sort of hypothetical Frankenstein ‘Texas hibernating bat’. I would recommend the authors take a more conservative approach and focus their attention on MYVE and PESU alone, and carefully point out which infection parameters are derived from MYLU and disease ecology in the NE, and to the extent possible, remove this source of unnecessary bias. If species level data are not available it’s understandable why the authors went with the route they did, so when MYLU parameters are used they should be contextualized for the reader as MYLU are not known to typically range in Texas. For example, how would differences in water vapor deficit (cave environment) and evaporative water loss (host) impact projections on fungal growth/shedding between the NE and Texas? The author’s method section was challenging to understand. I think it needs to first focus on carefully explaining the measured data and then describe how these data were modeled. The current organization jumps around and leads to unnecessary and avoidable reader confusion. I also was concerned that the detection effort in 2018 and 2020 is reported as unknown, which will introduce considerable bias into the parameterization ‘validation step’.

53 - ‘new habitable environmental reservoir pathogens’ - is unclear. Perhaps you mean something along the lines of ‘new habitat for pathogens found in environmental reservoirs’?

77 - would ‘population demographics’ be more appropriate than ‘population composition’?

89-92 - This is a long sentence that is a lot to take in for a reader. The last 2 chunks of ‘minimal information’ are presented without citations as to their importance, and their importance could use some further explanation. I.e. help the reader understand why you think the spatial layout of caves, which could mean either the spatial distribution of caves on the landscape or the spatial arrangement of microclimates and bats within a cave, relevant? Also why is the frequency of suitable caves relevant? As you point out in the next sentence, Pd is free living and can exist in environments absent bats.

90-91 - why the focus on caves here when certainly other hibernacula are present (culverts, buildings)? (confirmed below in the methods section)

107 - If this is the case, the Abstract is misleading at line 32, as the model is not species-specific.

123 - Concerningly, the Lilley et al. 2018 model was parameterized for MYLU (which are not found in Texas) and for WNS occurrence conditions in the Northeast U.S.. Methods provide no indication that host-associated parameters were adapted for Texan hibernating bat species (i.e. direct transmission rate, infection rate, carrying capacity, recovery rates, population growth rate) nor the appropriateness of applying this model to describe WNS threats in Texas. However, in line 199 it becomes clear the model was somewhat adapted to hibernating bats in Texas. A careful reorganization of the method section would provide more clarity to the reader.

135 - Unclear - you had environmental data from 94 counties or for 4251 caves? Thereafter unclear how measured environmental data from caves is used as the McClure et al. 2020 model is described as being used to predict mean cave temps for the geographic centers of each county. More confusion is added once I reached line 191 where collection of cave temperature data is described. Even more confusion is added in Supplement 1 where equations used to calculate ambient conditions are provided (S1 and S2). Perhaps this section of the manuscript can be reorganized so the reader is introduced first to the data collected and then how the model uses it.

137 - What is the temporal and spatial resolution of the temperature data used to calculate cave mean temp?

140 - Should clarify if the reference is for the approach used in McClure et al. 2020 or if the model from McClure et al. 2020 is being used

146 - Unclear how total number of patches were arrived at with binning etc.

150 - PRISM data needs a citation

153 - Not accounting for species structure seems questionable when WNS is only documented in MYVE in Texas along with others that are positive for Pd by not WNS.

166 - Notation is difficult to read, looks like it needs some extra spacing -

167 - Any data to support MYVE uses torpor at temps up to 12.5°C?

175 - The sigmoidal infectivity does have notable effects on disease dynamics - as shown in the Supplement 1. Given their huge effects (at 10 years the sigmoid-effect estimated there would be ~30 affected counties compared to 70 to 80 counties with the linear-effect) it's worthy of another mention in the results to give a range of potential outcomes.

201 - The parameter values came from which species specifically? When parameters were not available for Texas bat species, did you default back to parameters for MYLU?

202 - Only one hibernating bat species is reported with WNS in Texas, MYVE, so why not just model this species and PESU for a more conservative approach (and any other confirmed WNS species found in Texas I'm overlooking). This will have implications on the bat mortality estimates - which may likely be overestimates.

211 - How extensively was Texas surveyed in 2018 and 2020? If the survey in 2018 was partial would that introduce bias to the approximated values used in the 5-19 year modeling steps? This is further concerning as line 230 indicates the authors had no information on the actual detection effort or efficiency.

236 - Something unclear here - 'and were not found by this validation step'?

247 - needs 'counties' after mid Texas?

249 - 'rich' used again - repetitive to reader

274 - Very concerning that the model is most sensitive to changes in direct transmission rate and infection rate - that are pulled from studies of different bat species especially as we know there are interspecific differences in these parameters. Not mentioned here - importantly it also appears sensitive to amount of free-living fungus in initially affected counties - which was approximated and speaks to my concern about detection effort and bias. It was also sensitive to bat pop growth rate (which was based on MYLU).

275 - Much too long of a sentence for readers.

281 - The contributions of humidity to WNS susceptibility/outcomes is overlooked and should be brought into the discussion to further contextualize the discussion of temperature results.

296 - The finding of WNS in central Texas in the model is not at all surprising because it was parameterized using data from 2018-2020 and WNS was already detected in central Texas in 2020 - so the model is not actually making a new prediction about where WNS will be found in Texas.

336 - Needs grammatical work on the comparative if you want to use 'than' i.e. longer periods of torpor...

344 - Agree completely - including species specific variation!

348 – The model is focused on Texas so I would temper this statement as geographies outside of Texas may also contribute to the source-sink dynamics. Also how does this square with the early circa 2006-2010 Pd spread basically occurring from north to south (opposite of the direction you propose)? Does the lack of karst habitat across vast sections of the mid-west speak to the anticipated dynamics?

358 – Though again these are for MYLU

360 – Does 2 years of pathogen spread with unknown detection effort really provide a solid baseline for parameterizing this type of model?

374 – Source-sink dynamics are more about where the pathogen spreads from and to, it doesn't necessarily speak to susceptibility and potential for local extinctions, which is more about temp/humidity conditions and at what temperatures different species use torpor.

379 – Why is movement data the best way to estimate infectiousness? Please expand. I'm wondering about this conclusion as I interpret infectiousness here to be concerned with understanding aspects of direct transmission. Given the finding that 90% of the force of infection comes from environmental transmission, environmental measures of temp/humidity will likely give you the best indication of Pd suitability, and the observation that Pd is already found in the north and central regions of Texas how will this knowledge improve management decisions. To add to the last argument, the model is also predicting pretty much every population will have fungal spores in 5-10 years (Appendix S10). Also surprisingly no mention of looking at measuring species-specific parameters for further targeting efforts for future work?

576 – Mathematical notation issues in the creation of the pdf I suspect

587 – I think you mean 2018?

Supplement 1

- Stick/check with British or American spellings throughout (parameterization vs. parameterisation)
- Please address '(ref Anttila)'
- Standardize use of your Pd abbreviation throughout (i.e. not pd)
- 'initially affected patches (2017)' – wasn't it based on 2018 data?
- Figure titles shouldn't include variable symbols that require a new reader to visit a table to translate, rather use the parameter name itself.
- S8-S9 – consider revisiting your color gradient so it is more informative for this figure (everything looks vaguely yellow)
- S10 – is a helpful figure for readers – consider bringing this into the main narrative as a figure!

Supplement 2

2 Data – provide resolution spatial and temporal of the PRISM data you use

2.3 How was the filter for 'winter' duration, i.e. hibernation period determined? Perhaps provide a reference from roost acoustic activity?

Generally confused as the main text references McClure et al. 2020 as the source of the model for temperature data for caves – but after reviewing the code, it seems a fix might be to clarify the reference in the main text is using the approach used in McClure et al. 2020 and not that specific model.

Needs a spellcheck – see 'therefor'

Appendix S7

Can the authors provide some context on what range of sigmoid crossing point values are applicable?

Author's Response to Decision Letter for (RSPB-2020-2901.R0)

See Appendix B.

RSPB-2021-0719.R0

Review form: Reviewer 1

Recommendation

Accept as is

Scientific importance: Is the manuscript an original and important contribution to its field?

Good

General interest: Is the paper of sufficient general interest?

Good

Quality of the paper: Is the overall quality of the paper suitable?

Good

Is the length of the paper justified?

Yes

Should the paper be seen by a specialist statistical reviewer?

No

Do you have any concerns about statistical analyses in this paper? If so, please specify them explicitly in your report.

No

It is a condition of publication that authors make their supporting data, code and materials available - either as supplementary material or hosted in an external repository. Please rate, if applicable, the supporting data on the following criteria.

Is it accessible?

Yes

Is it clear?

Yes

Is it adequate?

Yes

Do you have any ethical concerns with this paper?

No

Comments to the Author

I am happy with the comments and changes made to the manuscript. I feel that the authors spent time on reorganizing the paper to make the model development and parameterize clearer to the reader. Though the model is still complex, I feel that the manuscript now explains it better enough so that the results can be correctly interpreted.

Decision letter (RSPB-2021-0719.R0)

05-May-2021

Dear Dr Meierhofer

I am pleased to inform you that your Review manuscript RSPB-2021-0719 entitled "Ten-year projection of white-nose syndrome disease dynamics at the southern leading-edge of infection in North America" has been accepted for publication in Proceedings B.

The referee and the Associate Editor do not recommend any further changes. Therefore, please proof-read your manuscript carefully and upload your final files for publication. Because the schedule for publication is very tight, it is a condition of publication that you submit the revised version of your manuscript within 7 days. If you do not think you will be able to meet this date please let me know immediately.

To upload your manuscript, log into <http://mc.manuscriptcentral.com/prsb> and enter your Author Centre, where you will find your manuscript title listed under "Manuscripts with Decisions." Under "Actions," click on "Create a Revision." Your manuscript number has been appended to denote a revision.

You will be unable to make your revisions on the originally submitted version of the manuscript. Instead, upload a new version through your Author Centre.

1) A text file of the manuscript (doc, txt, rtf or tex), including the references, tables (including captions) and figure captions. Please remove any tracked changes from the text before submission. PDF files are not an accepted format for the "Main Document".

2) A separate electronic file of each figure (tiff, EPS or print-quality PDF preferred). The format should be produced directly from original creation package, or original software format. Please note that PowerPoint files are not accepted.

3) Electronic supplementary material: this should be contained in a separate file from the main text and the file name should contain the author's name and journal name, e.g. `authorname_procb_ESM_figures.pdf`

All supplementary materials accompanying an accepted article will be treated as in their final form. They will be published alongside the paper on the journal website and posted on the online figshare repository. Files on figshare will be made available approximately one week before the accompanying article so that the supplementary material can be attributed a unique DOI. Please see: <https://royalsociety.org/journals/authors/author-guidelines/>

4) Data-Sharing and data citation

It is a condition of publication that data supporting your paper are made available. Data should be made available either in the electronic supplementary material or through an appropriate repository. Details of how to access data should be included in your paper. Please see <https://royalsociety.org/journals/ethics-policies/data-sharing-mining/> for more details.

<http://datadryad.org/submit?journalID=RSPB&manu=RSPB-2021-0719> which will take you to your unique entry in the Dryad repository.

Once again, thank you for submitting your manuscript to Proceedings B and I look forward to receiving your final version. If you have any questions at all, please do not hesitate to get in touch.

Sincerely,
Professor Hans Heesterbeek
<mailto:proceedingsb@royalsociety.org>

Associate Editor

Comments to Author:

Thanks for paying close attention to the reviewer suggestions in making the new version of the manuscript.

Reviewer(s)' Comments to Author:

Referee: 1

Comments to the Author(s).

I am happy with the comments and changes made to the manuscript. I feel that the authors spent time on reorganizing the paper to make the model development and parameterize clearer to the reader. Though the model is still complex, I feel that the manuscript now explains it better enough so that the results can be correctly interpreted.

Decision letter (RSPB-2021-0719.R1)

11-May-2021

Dear Dr Meierhofer

I am pleased to inform you that your manuscript entitled "Ten-year projection of white-nose syndrome disease dynamics at the southern leading-edge of infection in North America" has been accepted for publication in Proceedings B.

Data Accessibility section

Open Access

Paper charges

Sincerely,

Proceedings B

Appendix A

The paper uses an ordinary differential equation system to model a metapopulation of bats suffering from white-nose syndrome as they spread throughout Texas, USA. This work is a natural extension of previous models on this application and is sorely needed considering the urgency of the WNS situation. This paper is very well-written and organized, its ecology and mathematics seem appropriately applied and its conclusions fit with their findings. I have just two concerns about the paper, specifically about the sensitivity analysis.

1) On lines 225-231 you give the threshold values for the rejection criteria you used for your posterior distribution of your parameter sampling. You note that the thresholds were arbitrary, which is fine, but you should also investigate if your results are sensitive to the rejection criteria. Depending on computational requirements, you should try looking at a large range of thresholds and determine if the conclusions change based on reasonable variations in the thresholds.

2) In general, the sensitivity analysis conducted in Supplement 1 appears is a local sensitivity analysis (varying one parameter at a time) as opposed to a global sensitivity analysis (varying all parameters simultaneously). Global sensitivity analysis is far more accurate, especially in systems biology when uncertainty in models is common.

If the authors have solid reasoning to not employ a global sensitivity analysis, it should be stated in the paper. One such example would be computational limitations. But if they stick with a local sensitivity analysis, then conclusions from the sensitivity analysis should be made with caution.

If the authors are unfamiliar with global sensitivity analysis, then I suggest the following paper to learn about such approaches: Simeone Marino, Ian B. Hogue, Christian J. Ray, Denise E. Kirschner,

A methodology for performing global uncertainty and sensitivity analysis in systems biology,

Journal of Theoretical Biology,

Volume 254, Issue 1,

2008,

Pages 178-196,

ISSN 0022-5193,

<https://doi.org/10.1016/j.jtbi.2008.04.011>

Appendix B

Dear Dr Meierhofer:

I am writing to inform you that your manuscript RSPB-2020-2901 entitled "Ten-year projection of white-nose syndrome disease dynamics at the southern leading-edge of infection in North America" has, in its current form, been rejected for publication in Proceedings B.

This action has been taken on the advice of referees, who have recommended that substantial revisions are necessary. With this in mind we would be happy to consider a resubmission, provided the comments of the referees are fully addressed. However please note that this is not a provisional acceptance.

Sincerely,

Professor Hans Heesterbeek
mailto:proceedingsb@royalsociety.org

Associate Editor

Board Member: 1

Comments to Author:

This manuscript tackles an important topic and the referees uniformly support the efforts and approach.

However, they make detailed and critical comments especially regarding the choices around model and parameterization. In developing the next version of the manuscript, I think it is critical that the authors address:

- (i) clarity, so that readers can follow what is being done with the model to generate the results;
- (ii) justification of model assumptions (especially the one-population approximation - I think that details on how multiple species interactions might affect this should be given) and how parameters were selected/estimated (again, acknowledging that species/locations may have different parameters);
- (iii) furthermore, since there are parameters here that are hard to estimate or questionable, performing a global sensitivity analysis would partially allay concerns around the sensitivity of the model to these assumptions/estimates.

.....

Reviewer(s)' Comments to Author:

Referee: 1

Comments to the Author(s)

The paper “Ten-year projection of white-nose syndrome disease dynamics at the southern leading-edge of infection in North America” is a well thought-out manuscript that fits the scope of the journal and meets the journal's standards of originality, quality and language. The authors present a means of predicting white-nose syndrome spread based on cave densities and average dispersal distances of bat species, in addition to predicting the impacts of WNS on bat populations in Texas. Though potential impacts of WNS on populations has been modeled across the United States, specifically taking into account the spatial variation in environmental conditions that impact the causal fungus is timely. The work is interesting and will add a lot of important knowledge to management of this devastating disease. There were some spots of the manuscript that require clarification, but I don't want these issues to take away from the value of this work nor the appropriateness of the modeling approach.

My one main concern is of the complexity of the modeling effort and the ability for this approach to be described well enough for the reader to understand. I applaud the authors for trying to tackle these source-sink disease dynamics in a cave system with a hard-to-track animal, but I do think there needs to be some more clarity in how the model is presented, applied, and parameterized. I was left confused in how all of the many pieces fit together and how the many parameters were defined. I actually was a peer reviewer on the Lilley et al. (2018) manuscript, and I recall having a hard time wrapping my head around the modeling approach. Therefore I am unable to comment on the validity of the results presented here, as I was unsure how they were obtained. Additionally, there should be some careful consideration in how these results are discussed in the discussion: these are predictions, based on simplifications of a system using

parameters averaged over multiple species and locations. Therefore the authors should take care in making broad statements in the discussion given these results. Instead the authors should note the predicted declines rather than taking these results as hard truths.

I understand why the methods were set up in terms of describing the model first and then the parameterization, but it makes it difficult to go back and forth between how the parameters are used and how they were measured. I suggest reorganizing this entire section into sub-sections based on each model component, where the model component can be described and then discussed where the data come from the parameterization of that component.

- We thank the reviewer for their comprehensive assessment of our manuscript. We agree with the fact that our issues lie in how we described the methods and that our discussion of results may have overstepped. As such, we have taken care to adjust our discussion to acknowledge that our results are from a model and are not necessarily what will happen. Further, we have completely reorganized the methodology section to ensure clarity and reduce overall confusion as to how parameters were applied within the model. We also provide a better discussion of the model within the methods.

Below I outline the line-specific comments that could hopefully clarify how the model is applied in this system.

Line by line comments:

Key words: write out white-nose syndrome; may be also appropriate to include *Pseudogymnoascus destructans*.

- We have written out 'white-nose syndrome'. Unfortunately, there is a limit of 6 key words and so we are unable to add *P. destructans*

Line 48: Since there are so many other abbreviations in the manuscript, and EID is only referenced 3 times, it would be more appropriate to not use the abbreviation.

- We have replaced 'EID' with 'Emerging infectious diseases'

Line 56: And host physiology, such as the case with WNS.

- We have included 'host physiology'

Line 58: and behavior

- We have included 'behavior'

Line 63: mortality of the host

- We have added 'of the host'

Line 78: Pet peeve: though WNS itself is not what is being spread, but rather the fungus which then causes the disease. Change “WNS disease dynamics spread” to “fungal spread”

- We appreciate the reviewer for making this point. We have made the recommended change.

Line 86: Break up into two separate sentences instead of having the comma between the United States and many

- We have broken up the sentence into two separate sentences.

Line 88: again, WNS is not invading, the fungus is. Change “infection” to “observation”

- We have changed ‘infection’ to ‘observation’

Line 103: clarify from what the demographics and environmental data referenced here are measured from (i.e. bat demographics and hibernacula environmental data)

- We have clarified that we are using bat demographics and hibernacula environmental data.

Line 109: Please clarify this hypothesis: by stating “high cave concentrations” do you mean high concentrations of caves in an area? High concentrations of bats in a cave? And by stating “bat abundance,” do you mean bat abundance across the landscape? Bat abundance within the cave (and therefore density)? And on this thought, do you consider how cave size can therefore affect dynamics regarding bat abundance – the same number of bats within a small cave vs. a larger cave should impact fungal spread.

- We have clarified that we mean high concentrations of caves in an area and bat abundance across the landscape. We did not include cave size within our model, but we understand that size, as well as other aspects of caves (e.g., cave portal direction, microclimate availability) influence use of caves by bats. Generally, access to these data are difficult. If we were able to obtain it, adding these data into our model would add more dimensionality, complicating the results.

Line 111: Should ecology really be physiology here?

- We have changed ‘ecology’ to ‘physiology’

Line 111: It would be helpful to see some predictions following the hypotheses. This would allow the reader to know exactly what the paper is testing to confirm or falsify these hypotheses.

- We have included predictions following our hypotheses.

Line 123-126: Since the model is not described within the text, and the descriptions in the Supp Materials are mathematically dense, it may be worthwhile to include a flow chart that shows the different components of the model (such as in Figure 1 in Lilley et al. 2018). Specifically, what is modified from the Lilley et al. (2018) model that is then applied in this manuscript?

- We have modified the figure from Lilley et al. 2018 and included it within the supplementary documents. The main change from the original model to the one in the current manuscript is the lack of microbiome. As the modifications were minimal, we felt that including it within the manuscript was not necessary.

Line 131: thank you for including your GitHub repository!

- We thank the reviewer for acknowledging our GitHub repository.

Line 135: TSS not needed

- We have removed ‘TSS’

Line 137: Why were the sites grouped into temperature bins? How is that applied later on? It is not clear here.

- The binning was done to reduce the number of patches considered in the simulations. We originally tried to consider each populated hibernaculum as a patch, but quickly realized that this would not be feasible due to computational effort required to deal with such a large system. We therefore decided to do binning based on the mean temperature for hibernacula that would be otherwise identical. We have explained this more clearly.

Line 140: thank you for including your code.

- We thank the reviewer for acknowledging our code.

Line 143: I see now that the temperature bins are the “patches” – but these are not “patches” in a geographic sense, correct? Would distance between caves therefore influence spread?

- This is correct, each model patch corresponds to one or more geographical hibernaculum. Not that our data yield the hibernaculum count for each county but not their locations within the county. We consider within-county migration following the expected distance between randomly placed points inside the county geometry, and between-county migration following the distances between county midpoints. The number of affected patches is sensitive to distances between caves.

Line 145-146: How was this 40% applied across patches? And what survey data?

- We have clarified that the 40% was applied evenly across patches. We actually used the 40% of the hibernation sites prior to the binning to get the patches. We have cited my

unpublished data from four years of winter bat survey work (2015–2019) in Texas. These data included counts of bats by species in all potential hibernacula accessed across the state. These data are unpublished as reports were submitted to Texas Parks and Wildlife, and will not be released to the public until two years after the completion of the project.

Line 146: Patch number should be moved to results

- We have moved the sentence on patch number to the results.

Lines 154-157: Could this be represented in a figure to visualize these parameters?

- We considered how we could represent this as a figure to visualize the parameters. However, we could not determine a good visual for this that would clarify confusion.

Lines 269-273: this sounds like discussion rather than just reporting results

- We have moved this section on direct transmission of the fungus into the discussion section where appropriate.

Table 1: It is unclear if each of the “approximated” parameters were approximated in this manuscript or elsewhere. Perhaps either add more detail to the approximation method in the table, or outline the methods section to match the different parameters in the table to the reader can reference the text easily.

- We have added more information about the approximated values into the table description. We have also modified the methods section to provide more clarity to the reader.

Figure 1: This figure is hard to interpret. Either the figure needs to be clearer in its presentation and/or the legend needs to be more specific in how to interpret.

- We have attempted to clarify the figure description for a better understanding.

Referee: 2

Comments to the Author(s)

Please see attached file.

The paper uses an ordinary differential equation system to model a metapopulation of bats suffering from white-nose syndrome as they spread throughout Texas, USA. This work is a natural extension of previous models on this application and is sorely needed considering the urgency of the WNS situation.

This paper is very well-written and organized, its ecology and mathematics seem appropriately applied and its conclusions fit with their findings. I have just two concerns about the paper, specifically about the sensitivity analysis.

1) On lines 225-231 you give the threshold values for the rejection criteria you used for your posterior distribution of your parameter sampling. You note that the thresholds were arbitrary, which is fine, but you should also investigate if your results are sensitive to the rejection criteria. Depending on computational requirements, you should try looking at a large range of thresholds and determine if the conclusions change based on reasonable variations in the thresholds.

- Reviewer is correct, these thresholds are indeed largely arbitrary. We have now investigated the sensitivity to local variations of the threshold parameters. The range we have investigated is still relatively limited because changing the parameters changes the acceptance rate and may reduce drastically the number of simulations from which we calculate posterior means. Stepwise changes in the set of accepted simulations also causes some wobble to the resulting sensitivity curves, but altogether we can easily see which parameters are most affected by our chosen threshold values.

2) In general, the sensitivity analysis conducted in Supplement 1 appears is a local sensitivity analysis (varying one parameter at a time) as opposed to a global sensitivity analysis (varying all parameters simultaneously). Global sensitivity analysis is far more accurate, especially in systems biology when uncertainty in models is common.

If the authors have solid reasoning to not employ a global sensitivity analysis, it should be stated in the paper. One such example would be computational limitations. But if they stick with a local sensitivity analysis, then conclusions from the sensitivity analysis should be made with caution.

If the authors are unfamiliar with global sensitivity analysis, then I suggest the following paper to learn about such approaches: Simeone Marino, Ian B. Hogue, Christian J. Ray, Denise E. Kirschner, A methodology for performing global uncertainty and sensitivity analysis in systems biology, *Journal of Theoretical Biology*, Volume 254, Issue 1, 2008, Pages 178-196, ISSN 0022-5193, <https://doi.org/10.1016/j.jtbi.2008.04.011>

- As noted by the referee, a global sensitivity analysis would require considerable computational effort, we do not consider it feasible at this point. Furthermore, communicating and interpreting the outcomes under simultaneously varied parameters is challenging, especially since we summarise the outcome in more than one dependent variable (population reduction and number of affected patches).

Referee: 3

Comments to the Author(s)

I think the authors did an exceptional job working with the data and models available to make 5 and 10-year projections of Pd spread and WNS mortality for Texas. Clearly, models like this are heavily dependent on initial conditions and underlying assumptions. So the question becomes,

are the assumptions and data on initial conditions sufficient to support the results and conclusions? In particular my main concerns are, infectious parameters appear drawn from a species not found in Texas (MYLU), all hibernating bat species are modeled as one population – when we know there are inter-specific responses to Pd and striking inter-specific differences of WNS mortality among hibernating bats, and several parameters were derived from a validation exercise that had no data on surveillance/detection effort. The methods section is currently poorly written and jumps around. Careful revisions to this section may help alleviate these concerns. More explicit recognition of which parameters are pulled from MYLU and how these may or may not be applicable to an amalgam of all Texas hibernating bats is also needed. Overall these issues leave me with deep concerns about the validity of the results and conclusions. I'm keen on the premise, I'm just not convinced the data available are sufficient for the task at hand.

The authors made a choice, due I suspect to lack of species-level data, to model all hibernating bats as one population, which ignores known inter-specific threats from WNS. It also fails to mention that only one species, MYVE appears to actually get WNS in Texas (as of 2020), though I think the community is waiting to see what happens to PESU. Also unclear are which 'species' contributed parameter values that were then lumped together in this analysis, in a sort of hypothetical Frankenstein 'Texas hibernating bat'. I would recommend the authors take a more conservative approach and focus their attention on MYVE and PESU alone, and carefully point out which infection parameters are derived from MYLU and disease ecology in the NE, and to the extent possible, remove this source of unnecessary bias. If species level data are not available it's understandable why the authors went with the route they did, so when MYLU parameters are used they should be contextualized for the reader as MYLU are not known to typically range in Texas. For example, how would differences in water vapor deficit (cave environment) and evaporative water loss (host) impact projections on fungal growth/shedding between the NE and Texas? The author's method section was challenging to understand. I think it needs to first focus on carefully explaining the measured data and then describe how these data were modeled. The current organization jumps around and leads to unnecessary and avoidable reader confusion. I also was concerned that the detection effort in 2018 and 2020 is reported as unknown, which will introduce considerable bias into the parameterization 'validation step'.

- We appreciate the reviewer for their thoughtful feedback of our manuscript. We understand that they are concerned with the results of the model based on the data available and assumptions. Indeed, we have done the best that we could with what was available at the time we developed the model. We have described the caveats and included the model code so that anybody can run the model with their own parameter set. The outcomes of our model are just one vision of what the situation could turn out like in Texas.

We have further cleared up some confusion about bat species; we understand that only one species in Texas has WNS, and that some parameters stem from species not known to Texas. We have clarified this within the text. Unfortunately, again, many parameter values are just not available as not mentioned within the manuscript. As such, we mentioned that we decided to model combine available information on cave-hibernating species as a result of the lack of species-specific data. Of importance, the parameters that incorporated a species outside of Texas (*M. lucifugus*) pertain to infection rate and recovery rate, which are now indicated within our table of parameters. All parameters were averaged values based on known information of species with ranges in Texas. We believe that we have now clearly stated this within the manuscript so

as to be as transparent as possible. We further believe that although this is not a species-specific model, the model still provides an idea of what may happen in Texas. Further, as more information becomes available, our model can be used to give more species-specific predictions.

Given the confusion with the methods stressed by the reviewer, we have greatly modified the entire section to ensure less confusion for readers. We start with discussing model development, followed by spatial setting, and ending with parameters and parameterization. We believe this is the best way to lay out our methods section.

53 – ‘new habitable environmental reservoir pathogens’ – is unclear. Perhaps you mean something along the lines of ‘new habitat for pathogens found in environmental reservoirs’?

- We have clarified the sentence with the suggestion of the reviewer.

77 – would ‘population demographics’ be more appropriate than ‘population composition’?

- We have changed ‘population composition’ to ‘population demographics’

89-92 – This is a long sentence that is a lot to take in for a reader. The last 2 chunks of ‘minimal information’ are presented without citations as to their importance, and their importance could use some further explanation. I.e. help the reader understand why you think the spatial layout of caves, which could mean either the spatial distribution of caves on the landscape or the spatial arrangement of microclimates and bats within a cave, relevant? Also why is the frequency of suitable caves relevant? As you point out in the next sentence, Pd is free living and can exist in environments absent bats.

- We have added in the Lilley et al. 2018 citation for our statement on cave distribution and frequency to highlight the importance of including these factors within our model. We have further added some reasoning to why cave frequency can increase potential of infection. Specifically, we added that Pd is free living and can exist in environments in the absence of bats, and as such, a high frequency of caves provides more potential of infection.

90-91 – why the focus on caves here when certainly other hibernacula are present (culverts, buildings)? (confirmed below in the methods section)

- We have added a brief justification for our use of caves as opposed to other hibernacula within in introduction.

107 – If this is the case, the Abstract is misleading at line 32, as the model is not species-specific.

- We have changed ‘species’ to ‘hosts’.

123 – Concerningly, the Lilley et al. 2018 model was parameterized for MYLU (which are

not found in Texas) and for WNS occurrence conditions in the Northeast U.S.. Methods provide no indication that host-associated parameters were adapted for Texan hibernating bat species (i.e. direct transmission rate, infection rate, carrying capacity, recovery rates, population growth rate) nor the appropriateness of applying this model to describe WNS threats in Texas. However, in line 199 it becomes clear the model was somewhat adapted to hibernating bats in Texas. A careful reorganization of the method section would provide more clarity to the reader.

- We have clarified where the parameters come from both within text as well as in the table description. We have also clarified that unfortunately a lot of the data for parameters are not readily available nor would be accurate, and as such, we are not modeling one species. Although a couple parameters stem from a species not known to Texas, which we have clarified in text, we feel that this is a good starting point until more data are gathered and tested with our model. As the methods were identified as a source of confusion, we have reorganized for clarity.

135 – Unclear – you had environmental data from 94 counties or for 4251 caves? Thereafter unclear how measured environmental data from caves is used as the McClure et al. 2020 model is described as being used to predict mean cave temps for the geographic centers of each county. More confusion is added once I reached line 191 where collection of cave temperature data is described. Even more confusion is added in Supplement 1 where equations used to calculate ambient conditions are provided (S1 and S2). Perhaps this section of the manuscript can be reorganized so the reader is introduced first to the data collected and then how the model uses it.

- We removed the part about the environmental data in this sentence as it was misinterpreted and did not fit within what was being stated. We obtained external environmental data for each Texas county and so our previously written sentence was misleading. As the previous statement, we have modified the structure of the methods section, which should help to clear up the confusion on where data are from and how they are entered into the model.

137 - What is the temporal and spatial resolution of the temperature data used to calculate cave mean temp?

- We have clarified that the spatial resolution of the external temperatures from PRISM is 4km grid cell resolution. The temporal resolution is daily within 2017 and is mentioned in the methods.

140 – Should clarify if the reference is for the approach used in McClure et al. 2020 or if the model from McClure et al. 2020 is being used

- We have clarified that we used the approach given in McClure et al. 2020.

146 – Unclear how total number of patches were arrived at with binning etc.

- In the beginning of the Results section we now write “In total, there were 4251 hibernation sites occupied by bats aggregated into 293 patches within our model.” I.e.

the binning reduces the number of patches we need to consider in the spatial model. Within each county, we aggregated several hibernation sites to a patch according to 2 degree C bins of a gaussian distribution of cave temperatures.

150 – PRISM data needs a citation

- We have added the citation for PRISM data.

153 – Not accounting for species structure seems questionable when WNS is only documented in MYVE in Texas along with others that are positive for Pd by not WNS.

- We did not account for species structure as we are modeling all cave hibernating species as one population to give a simplistic overview of what may happen, which we've discussed within the manuscript. There may be bat species that have WNS that have not yet been encountered yet through current survey efforts. Further, a lot of interactions between species are not yet known, and our model attempts not to disclose these.

166 – Notation is difficult to read, looks like it needs some extra spacing

- We adjusted the spacing.

167 – Any data to support MYVE uses torpor at temps up to 12.5°C?

- Previous work conducted in Texas suggests that cave myotis have torpid skin temperatures reaching that temperature. See Meierhofer et al. 2019: Winter Habitats of Bats in Texas, <https://doi.org/10.1371/journal.pone.0220839>. We have cited that manuscript which described skin temperatures and roost temperatures of seven bat species in Texas. Specific to cave myotis, skin temperatures during winter were an average of 12.41 ± 4.05 . These temperatures were further correlated with the substrate temperature to provide further support that bats were hibernating when temperatures were taken. Thus, we believe that 12.5 °C is an acceptable temperature for our model. Again, our model is not to be taken at a species-specific level, but we did attempt to the best of our ability to have values that broadly represent hibernating bat species in Texas.

175 – The sigmoidal infectivity does have notable effects on disease dynamics – as shown in the Supplement 1. Given their huge effects (at 10 years the sigmoid-effect estimated there would be ~30 affected counties compared to 70 to 80 counties with the linear-effect) it's worthy of another mention in the results to give a range of potential outcomes.

- We have provided a brief mention about how under sigmoidal infectivity response, the range of potential outcomes is wider. Depending on parameterization, after 10 years, we expect the number of patches (counties) affected to be 70-80.

201 – The parameter values came from which species specifically? When parameters were not available for Texas bat species, did you default back to parameters for MYLU?

- We have attempted to clarify the confusion as to where parameter values come from both in text and within the description of the table. For the purpose of our model, we have

clearly stated that our values come from not one singular species, but from hibernating bat species found in Texas. In two circumstances, we used information from another species not found in Texas due to a lack of available information for species within Texas. These two parameters for *M. lucifugus* pertain to infection rate and recovery rate (noted in the table). We have clarified when this occurred. Our model is just one prediction that can be improved as more data become available, allowing for more species-specific analyses. We have identified this more clearly within text as well as in the table listing parameters.

202 – Only one hibernating bat species is reported with WNS in Texas, MYVE, so why not just model this species and PESU for a more conservative approach (and any other confirmed WNS species found in Texas I’m overlooking). This will have implications on the bat mortality estimates – which may likely be overestimates.

- The reviewer is correct; so far only one species has WNS. However, we anticipate that other hibernating species in Texas will get WNS. Again, values for parameters are not readily available to model a single-species, but we did our best to use values that broadly describe hibernating bat populations in Texas. When data are available, researchers can use our model with those new parameters and look at individual species.

211 – How extensively was Texas surveyed in 2018 and 2020? If the survey in 2018 was partial would that introduce bias to the approximated values used in the 5-19 year modeling steps? This is further concerning as line 230 indicates the authors had no information on the actual detection effort or efficiency.

- It would be difficult to do a complete survey of all caves in Texas and white-nose syndrome survey kits were limited. Over time, more caves were surveyed, but it is still just a sample of the total potential sites in Texas. We agree that any partial data would introduce bias and that is why we used this type of model. They provide the best estimate based on current knowledge, and can be improved later on when more data are available.

236 – Something unclear here – ‘and were not found by this validation step’?

- We have deleted this part of the sentence as it was a mistake.

247 – needs ‘counties’ after mid Texas?

- We have added ‘counties’

249 – ‘rich’ used again – repetitive to reader

- We have removed rich and reworded.

274 – Very concerning that the model is most sensitive to changes in direct transmission rate and infection rate – that are pulled from studies of different bat species especially as we know there are interspecific differences in these parameters. Not mentioned here – importantly it also appears sensitive to amount of free-living fungus in initially affected

counties – which was approximated and speaks to my concern about detection effort and bias. It was also sensitive to bat pop growth rate (which was based on MYLU).

- We have provided a discussion on the shortcomings of the model starting on line 343. Transmission rate is not going to be constant and will vary based on environment etc. As such these parameters cannot be known—only estimated. The parameterization looks at parameters all at once rather than one by one as the parameter values themselves may not have been accurately measured. The greatest challenge in reliability in the model is the fact that we only have two years. Given more time, a follow-up study can be conducted with more reliability.

275 - Much too long of a sentence for readers.

- We have broken down the sentence into multiple sentences for ease of interpretation.

281 – The contributions of humidity to WNS susceptibility/outcomes is overlooked and should be brought into the discussion to further contextualize the discussion of temperature results.

- We have added a brief discussion about the contributions of relative humidity to fungal growth and propagation. We also mention that bats also tend to hibernate at sites with high relative humidity, and that relative humidity affects hibernation success independently of *P. destructans*. We have further discussed how relative humidity data were not included within the model as the data were not available.

296 – The finding of WNS in central Texas in the model is not at all surprising because it was parameterized using data from 2018-2020 and WNS was already detected in central Texas in 2020 – so the model is not actually making a new prediction about where WNS will be found in Texas.

- We have removed the statement from the discussion as our model was indeed parameterized based on the initial status of fungal detection in the state and documentation of WNS.

336 – Needs grammatical work on the comparative if you want to use ‘than’ i.e. longer periods of torpor...

- We have changed out the word ‘extended’ for ‘longer’ to retain the comparison.

344 – Agree completely – including species specific variation!

- Thank you! We have also included ‘species-specific variation’ within the statement.

348 – The model is focused on Texas so I would temper this statement as geographies outside of Texas may also contribute to the source-sink dynamics. Also how does this square with the early circa 2006-2010 Pd spread basically occurring from north to south

(opposite of the direction you propose)? Does the lack of karst habitat across vast sections of the mid-west speak to the anticipated dynamics?

- Our apologies. In this sentence, we are actually referring to spread within Texas, not North America as whole. We have now clarified this in the text.

358 – Though again these are for MYLU

- We acknowledge that two of the parameters were sourced from a species not found in Texas and have noted this within text as well as in the table description. However, we were not trying to do a species-specific model given the lack of available species-specific data. We did the best we can with the information accessible. We consider it not to be important to get caught up on species level information, but rather see how well the model predicts with the values presented.

360 – Does 2 years of pathogen spread with unknown detection effort really provide a solid baseline for parameterizing this type of model?

- The robustness of our model and the sensitivity analyses suggest the model is good. However, we acknowledge that it will get better the more data is added. Therefore, we have provided all code and data used for this model, so that new data can be added at any stage to further improve the predictions (see line 352).

374 – Source-sink dynamics are more about where the pathogen spreads from and to, it doesn't necessarily speak to susceptibility and potential for local extinctions, which is more about temp/humidity conditions and at what temperatures different species use torpor.

-We have modified our statement to remove 'source-sink dynamics'

379 – Why is movement data the best way to estimate infectiousness? Please expand. I'm wondering about this conclusion as I interpret infectiousness here to be concerned with understanding aspects of direct transmission. Given the finding that 90% of the force of infection comes from environmental transmission, environmental measures of temp/humidity will likely give you the best indication of Pd suitability, and the observation that Pd is already found in the north and central regions of Texas how will this knowledge improve management decisions. To add to the last argument, the model is also predicting pretty much every population will have fungal spores in 5-10 years (Appendix S10). Also surprisingly no mention of looking at measuring species-specific parameters for further targeting efforts for future work?

- Thank you for this comment. After careful thought and based on our model output, we have deleted the statement. Movement is more important for transmission rather than infectiousness, and so the sentence was modified. The idea of looking specifically at species-specific parameters was completely overlooked, and as such, we have included

this as another suggestion for future research. We agree that this is important given information is lacking on several species, and thus building those models would be difficult if not impossible.

576 – Mathematical notation issues in the creation of the pdf I suspect

- We fixed the notation.

587 – I think you mean 2018?

- We have changed the date to ‘2018’ in both locations where it was mistyped.

Supplement 1

- Stick/check with British or American spellings throughout (parameterization vs. parameterization)

- We have addressed the spelling issues throughout.

- Please address ‘(ref Anttila)’

- We have now provided the reference for Anttila et al.

- Standardize use of your Pd abbreviation throughout (i.e. not pd)

- We have changed Pd to *P. destructans* to match the manuscript.

- ‘initially affected patches (2017)’ – wasn’t it based on 2018 data?

- We thank the author for catching our error. We have changed the date to 2018.

- Figure titles shouldn’t include variable symbols that require a new reader to visit a table to translate, rather use the parameter name itself.

- We have made appropriate changes to figure titles so that parameter names are now listed.

- S8-S9 – consider revisiting your color gradient so it is more informative for this figure (everything looks vaguely yellow)

- We considered the suggestion by the reviewer about changing the color gradient. However, these figures relate to figure 2 panels C and D within the manuscript. For comparison purposes, we used the same scale which is why the supplemental figure values all fall within the yellow color range of the scale. The difference in scaling is necessary, because bat loss predictions are so low in the two scenarios represented in the supplementary compared to the high predicted loss in the “full” scenario presented in the main text. The point of these figures was to demonstrate that loss due to Direct transmission alone < loss due to Environmental transmission alone < loss due to combined transmission. As such, we believe the original figures are more suitable.

- S10 – is a helpful figure for readers – consider bringing this into the main narrative as a figure!

- We thank the reviewer for the suggestion of moving the figure within text. However, given the page limits and all other recommended revisions, we have not moved the figure within the text.

Supplement 2

2 Data – provide resolution spatial and temporal of the PRISM data you use

- We have added the spatial and temporal resolution of the PRISM data used.

2.3 How was the filter for ‘winter’ duration, i.e. hibernation period determined? Perhaps provide a reference from roost acoustic activity?

- We have provided a citation to the Sandel et al. 2001 paper in which culvert hibernacula were surveyed monthly between September–March and count data reported. We have further cited Meierhofer et al. 2019 which reports roosting temperatures in relation to substrate temperatures of bats in Texas from November–early March to further support the months selected.

Generally confused as the main text references McClure et al. 2020 as the source of the model for temperature data for caves – but after reviewing the code, it seems a fix might be to clarify the reference in the main text is using the approach used in McClure et al. 2020 and not that specific model.

- We have clarified in-text that we used the approach used in McClure et al. 2020.

Needs a spellcheck – see ‘therefor’

- We have checked spelling throughout.

Appendix S7

Can the authors provide some context on what range of sigmoid crossing point values are applicable?

- This is unfortunately very challenging, and indeed one of the reasons we originally decided to use the linear response in the results presented in main text. The range of crossing points we study in the supplement is actually rather large (the units are in carrying capacity), and we explain this in the supplement.